# EXVERUS: Verus Proof Repair via Counterexample Reasoning

**Jun Yang** [1]   **Yuechun Sun** [1]   **Yi Wu** [1]   **Rodrigo Caridad** [1]   **Yongwei Yuan** [2]   **Jianan Yao** [3]   **Shan Lu** [1 4]   **Kexin Pei** [1]

## Abstract

Large Language Models (LLMs) have shown promising results in automating formal verification. However, existing approaches often treat the proof generation as a static, end-to-end prediction, relying on limited verifier feedback and lacking access to concrete instances of proof failure, i.e., *counterexamples*, to characterize the discrepancies between the intended behavior specified in the proof and the concrete executions of the code that can violate it. We present EXVERUS, a new framework that enables LLMs to generate and repair Verus proofs with actionable guidance based on the behavioral feedback using counterexamples. When a proof fails, EXVERUS automatically generates counterexamples, and then guides the LLM to learn from counterexamples and block them, incrementally fixing the verification failures. Our evaluation shows that EXVERUS substantially outperforms the state-of-the-art LLM-based proof generator in proof success rate, robustness, cost, and inference efficiency, across a variety of model families, agentic design, error types, and benchmarks with varying levels of difficulty.

## 1. Introduction

Large Language Models (LLMs) have shown promising results in formal verification, a task that uses rigorous mathematical modeling and proofs, written extensively by human experts, to ensure program correctness (Kozyrev et al., 2024; Song et al., 2024; First et al., 2023; Mugnier et al., 2025; Yang et al., 2025b; Chen et al., 2025; Aggarwal et al., 2025; Misu et al., 2024; Loughridge et al., 2025; Chakraborty et al., 2025; Wu et al., 2023; Sun et al., 2024a; Yan et al., 2025; Shefer et al., 2025). Automated proof generation has been widely accepted as an amenable task for LLMs, as the

unreliable outputs from LLMs can be formally checked by proof assistants and verifiers with provable guarantees. As a result, proof generation becomes a trial-and-error process, with feedback on proof failures guiding the LLM to repair the proof. This automated process makes formal methods more accessible to developers without specialized expertise.

Among existing verifiers, Verus (Lattuada et al., 2023; 2024) has been particularly amenable for developers to verify real-world systems (Zhou et al., 2024b; Sun et al., 2024b; Microsoft, 2024). Due to its Rust-native design, Verus allows developers to express their knowledge about safety and concurrency directly into proofs, making it practical to verify the correctness of large-scale, critical systems, including cluster management controllers (Sun et al., 2024b), virtual machine security modules (Zhou et al., 2024b), and microkernels (Chen et al., 2023).

Recent efforts in LLM-based Verus proof generation have primarily focused on prompting the LLM to generate proof annotations and iteratively repair verification failures based on verifier feedback (Zhong et al., 2025; Yang et al., 2025b; Yao et al., 2023; Aggarwal et al., 2025; Chen et al., 2025). However, these LLM-based approaches are largely constrained by static code patterns and error messages. The verifier error messages are often too coarse and ambiguous to reveal the root cause of the verification failure, e.g., `postcondition not satisfied`, lacking detailed elaboration needed to guide precise proof refinement.

To address this issue, existing techniques rely on expensive, handcrafted repair strategies as prompts for each error type (Yang et al., 2025b), or synthesizing datasets to enable large-scale training (Chen et al., 2025). The former suffers from the high cost of manual effort, and the handcrafted repair rules often fail to generalize to new error types and new Verus versions, while the latter incurs a nontrivial data curation cost, e.g., a month of non-stop GPT-4o invocations and rejection sampling (Chen et al., 2025).

**Actionable feedback - counterexamples.** In verification, traditional techniques frequently rely on counterexamples as strong guidance for debugging failures and refining proofs incrementally (Clarke et al., 2003; Bradley, 2012). Counterexamples serve as *witnesses* that ground abstract logical failures into specific, concrete states. By identifying a precise state where a proof fails, a counterexample acts as a

[1]The University of Chicago [2]Purdue University [3]The University of Toronto [4]Microsoft Research. Correspondence to: Kexin Pei <kpei@uchicago.edu>, Jun Yang <juny@uchicago.edu>.

*Proceedings of the 43rd International Conference on Machine Learning*, Seoul, South Korea. PMLR 306, 2026. Copyright 2026 by the author(s).

hard constraint to block the counterexamples and thus prune the search space of the proof. When combined with iterative counterexample-guided blocking, this transforms the open-ended, monolithic verification process into an incremental data-driven proof refinement workflow.

**Challenges in obtaining Verus counterexamples.** However, extracting semantically meaningful, actionable counterexamples directly from Verus' SMT backend is particularly challenging (Zhou et al., 2024a). First, Verus explicitly resolves key Rust semantics (e.g., ownership, borrowing, lifetimes, etc.) *before* generating low-level Verification Condition (VC) to produce smaller VCs for efficient solving. A lot of source-level semantic information is abstracted away. The lowering process exacerbates the problem by introducing extensive auxiliary artifacts, e.g., single static assignment (SSA) snapshots, without any direct mapping to the source program (Lattuada et al., 2023). Counterexamples are thus expressed over these lowered artifacts rather than over a faithful source-level state, making decompiling them into a readable and usable form often infeasible.

Second, Verus VCs heavily rely on quantifiers, e.g., `exists` and `forall`, but SMT solving with quantifiers is inherently incomplete. When faced with the monolithic, context-heavy queries produced by full-program VCs, the solver's fragile instantiation heuristics often return `unknown` or `time out`, and even successful counterexamples could be partial and fail to correspond to actual source-level executions (Zhou et al., 2024a).

**Our approach.** We present EXVERUS, a fully automated Verus proof generation framework guided by semantically meaningful, source-level counterexamples. Our key insight is to completely bypass the compilation of Verus proofs into massive, complex, low-level SMT queries and instead rely on the LLM to synthesize SMT queries that simulate the verification failure directly at the *source level*. Concretely, each synthesized query isolates the failing obligation and asks the solver for a concrete assignment to the original program variables that violates it, yielding concise, semantically meaningful counterexamples. Such counterexamples are better suited, as the proof is also written at the source level, using source-level variables and data structures.

Based on this insight, EXVERUS instructs the LLM to synthesize source-level SMT queries that efficiently search for counterexamples. Beyond faithfully translating proof annotations, the prompt asks the LLM to encode semantic information (e.g., type, data structures) into the naming convention of variables for source-level counterexample reconstruction. It also unleashes the creativity of LLMs to adaptively simplify the query by concretizing variables or avoiding quantifiers, e.g., assuming a concrete length for an arbitrary array `nums`, such that the burden of the solver is reduced while the validity of the counterexamples remains

checkable (Section 3.1).

Guided by these concrete, source-level counterexamples, EXVERUS can further summarize failure patterns, diagnose the root cause of the error, generalize from the error patterns to block them, and incrementally repair the proofs by iterating these steps. As the generated repair can always be validated by querying Verus, this entire process remains bounded, even when the correctness of counterexamples can occasionally be unverifiable, e.g., for non-inductive cases and sophisticated invariants.

**Results.** Our evaluation shows that EXVERUS substantially advances Verus proof repair in success rate, robustness, and cost efficiency. Across a wide variety of benchmarks, EXVERUS solves 38% more tasks on average than the state-of-the-art, and the advantage widens to $2\times$ on harder benchmarks such as LCBench and HumanEval. EXVERUS remains robust against obfuscated inputs under semantics-preserving transformations with success rates consistently above 73%, while the state-of-the-art stays below 50%.

## 2. Overview

### 2.1. Background: Automated Proof in Verus

In this work, we focus on Verus, a Rust-native tool for Rust code verification. Verus has been particularly appealing to developers working on verifying real-world systems (Zhou et al., 2024b; Sun et al., 2024b). Verus requires users to provide suitable *specifications*, e.g., *pre-conditions* and *post-conditions*, and *proof annotations*, e.g., *invariants* and *assertions*, to assist verification. The proof (including code, specifications, proof annotations) is processed by Verus to produce Verification Conditions (VCs) discharged to off-the-shelf satisfiability module theories (SMT) solvers for validity checking. For example, consider the following function that sums 1 to n.

```
1  fn sum_to_n(n: nat) -> (result: nat)
2      requires n >= 0,  // pre-condition
3      ensures result == n*(n+1)/2,  //
           post-condition
4  {
5      let mut i: nat = 0;
6      let mut sum: nat = 0;
7      while i < n
8          invariant  // proof annotation
9              sum == i*(i+1)/2,
10             i <= n,
11     {
12         i = i + 1;
13         sum = sum + i;
14     }
15     sum
16 }
```

The pre-condition, `n >= 0`, specifies the conditions that must be satisfied when the function is invoked. The post-condition, `result == n*(n+1)/2`, specifies the desired property after function execution, and is our proof

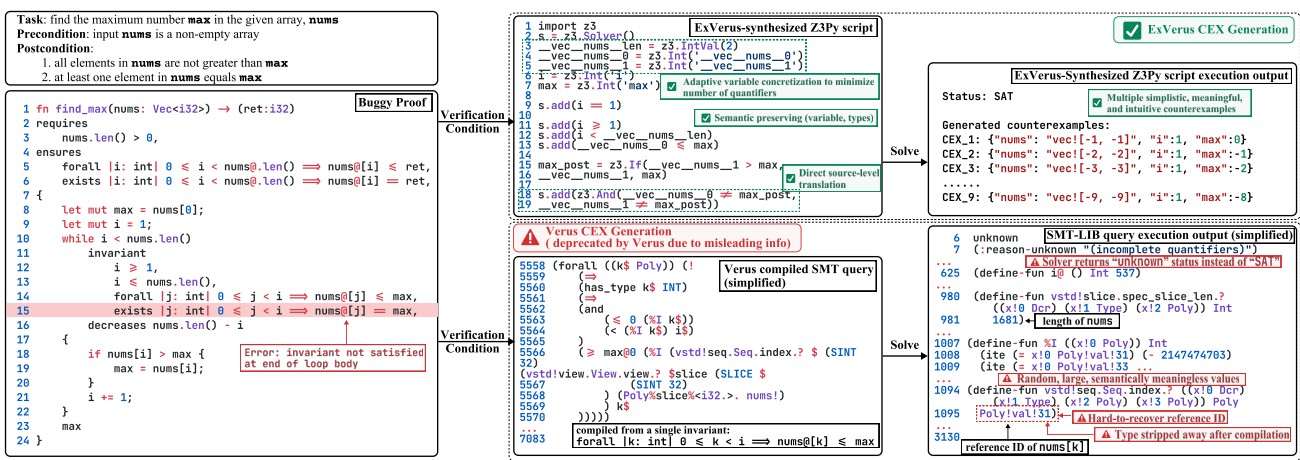

**Figure 1.** Motivating example from VerusBench (Misc/findmax) showing advantages of source-level counterexample (the EXVERUS counterexample generation on top right) vs. Verus's counterexample (Verus counterexample generation on the bottom right).

target. To complete the proof, the developer needs to provide proof annotations, in this case two loop invariants, `sum == i*(i+1)/2` and `i <= n`. These are properties that are true regardless of which iteration the loop is running at. Inferring such invariants has been a key barrier to automating formal verification (Flanagan & Leino, 2001; Garg et al., 2014; Kamath et al., 2023).

The existing LLM-based Verus proof generation approaches often adopt the paradigm of iteratively repairing verification failures based on verifier feedback, e.g., error messages (Zhong et al., 2025; Yang et al., 2025b; Aggarwal et al., 2025; Chen et al., 2025). However, due to lack of actionable feedback, e.g., the detailed information pinpointing the errors such as counterexamples, the error messages alone are often too coarse and ambiguous to reveal the root cause of the verification failure and guide the LLM to repair the proof. Therefore, they have to employ either finetuning (Chen et al., 2025) or heuristics-heavy, few-shot prompting (Yang et al., 2025b; Aggarwal et al., 2025; Zhong et al., 2025) to encode expert knowledge. The former incurs a nontrivial data curation cost, e.g., a month of non-stop GPT-4o invocations and rejection sampling (Chen et al., 2025), while the latter often fails to generalize to new error types and new versions.

**Counterexample-guided proof repair.** In formal verification, counterexamples have been used as a concrete, actionable feedback that effectively guides incremental proof synthesis and repair (Clarke et al., 2000; Bradley, 2011; Garg et al., 2014), because counterexamples precisely pinpoint the root cause of verification failures. However, generating counterexamples in Verus is particularly challenging. We use the following motivating example to describe these challenges, and motivate how EXVERUS's design attempts to address these challenges.

## 2.2. Motivating Example

Figure 1 illustrates the core challenges of using Verus' counterexample and the advantages of EXVERUS-generated counterexamples. The proof reports `invariant not satisfied at end of loop body`. This error message provides little evidence on why the invariant is not satisfied, e.g., whether the invariant is too weak or too strong, to effectively guide the repair. To diagnose this error, a user might try to extract a counterexample from the backend SMT solver output (bottom right), but this faces the following challenges.

When Verus compiles high-level Rust abstractions (e.g., `Vec<T>`, ghost code) into low-level SMT-LIB constraints, its lossy lowering strips semantic metadata (e.g., types, data-structure invariants) and introduces auxiliary artifacts (e.g., SSA snapshots) with no source-level counterpart (Lattuada et al., 2023). Recovering a faithful source-level state from such a low-level model is inherently undecidable without keeping nontrivial additional metadata.

In this example, Verus compiles the proof into a 7,083-line SMT query. Simply running the solver (Z3) to solve this query yields `unknown`[1] and a 3,130-line log (Figure 1 bottom-right). The Verus internal debugger reports `not implemented: assignments are unsupported in debugger mode`. So we have to manually inspect the log to recover the counterexample model. Even after the manual inspection, the counterexample remains noisy and hard to interpret: Z3 assigns a random, large value to `Poly!val!31`. A careful investigation indicates that it corresponds to `nums[k]` in the original code, but neither the value nor the reference ID has any semantic meaning. It also assigns `i=537` and

---

[1]Verus developers confirm that Verus frequently returns `unknown` due to its limited quantifier support (Verus Team, 2025).

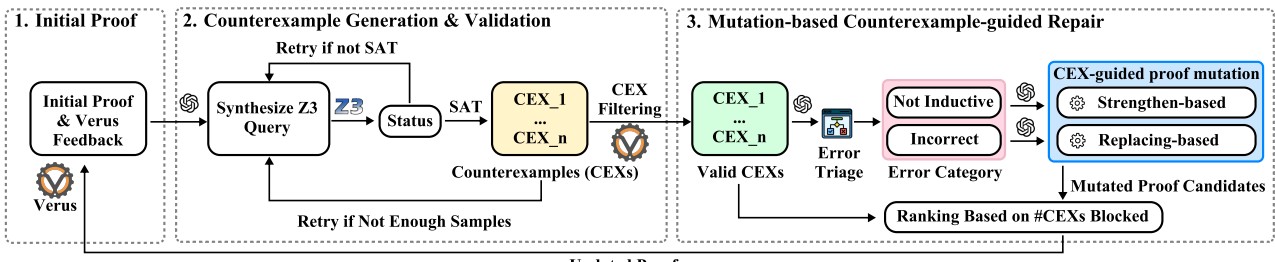

*Figure 2.* The EXVERUS Workflow. The framework operates in two core phases after the initial proof generation module. Firstly, the Counterexample Generation and Validation module leverages LLMs to synthesize source-level Z3Py queries, and executes the queries to produce concrete counterexamples that are subsequently validated by a dedicated verifier-based validation module. After that, the Mutation-based Repair module uses these validated counterexamples to guide error classification, apply customized mutations, and rank the mutated candidates to repair the proof.

`nums.len=1681` while leaving most elements unconstrained, providing little actionable guidance to repair, and could confuse users. In fact, Verus developers have decided to discontinue support for counterexamples due to the misleading values (see detailed discussions in Appendix F).

EXVERUS sidesteps these issues by *synthesizing counterexamples directly at the source level*. The top-right blocks show an EXVERUS-synthesized Z3Py script that concretizes the conditions that lead to the failed proof. It speculatively simplifies the assumption about the array length (e.g., `nums.len`=2), modeling only the first two elements. Solving this Z3Py script can quickly return concise, semantically meaningful counterexamples, e.g., `nums=vec![-1,-1]`, `i=1`, `max=0`. Because source-level names and types are preserved, the counterexample can be recovered in structured JSON, making it possible to replay and validate the counterexamples (Section 3.1). Further, EXVERUS can generate multiple counterexamples to make the generalizable failure patterns more salient.

Our key observation is that Verus' low-level models are hard to recover and interpret, while LLMs can generate concise, readable counterexamples, so it is more informative to help pinpoint the root cause of verification failures and elicit more actionable repair strategies.

### 2.3. Problem Formulation

We formally define the problem of counterexample-guided proof generation as an iterative optimization process. Given a program $\mathcal{P}$ with a specification $\Phi = (P_{pre}, Q_{post})$, i.e., pre-conditions and post-conditions, the task is to synthesize a proof $\Pi$ with a set of proof annotations (invariants, assertions, etc.) such that the program is provably correct. If, at step $t$, the proof has a single target verification error $e_t$, the goal of this step is to 1) generate a set of counterexamples that reveals $e_t$, and 2) mutate the proof to block the counterexamples to resolve $e_t$.

**Definition 2.1** (Counterexample). A counterexample $\sigma \in$

$\Sigma_t$ is a concrete program state that witnesses a verification failure in the current proof $\Pi_t$. For a failing verification constraint $A_t(\sigma) \implies C_t(\sigma)$ derived from $\Pi_t$, a valid counterexample satisfies:

$$\sigma \models A_t(\sigma) \land \neg C_t(\sigma) \tag{1}$$

where $A_t$ represents the antecedent (pre-state) and $C_t$ represents the consequent (post-state) at step $t$.

At each step $t$, there exists a set of counterexamples $\Sigma_t = \{\sigma_1, \ldots, \sigma_k\}$ that witness the failures of the current buggy proof $\Pi_t$. The objective is to generate an updated proof $\Pi_{t+1}$ that eliminates the counterexamples $\Sigma_t$, thus resolving the current verification failure.

**Definition 2.2** (Iterative Blocking). An updated proof $\Pi_{t+1}$ is a valid refinement relative to $\Sigma_t$ if it blocks all identified counterexamples. Formally, for every $\sigma \in \Sigma_t$, the updated verification constraint is no longer violated:

$$\forall \sigma \in \Sigma_t. \quad \sigma \not\models A_{t+1}(\sigma) \land \neg C_{t+1}(\sigma) \tag{2}$$

The process terminates when all verification failures are resolved (i.e., no counterexamples exist).

## 3. EXVERUS Framework

Figure 2 shows the high-level workflow of EXVERUS. It starts by taking as input a Rust program $\mathcal{P}$ and its specifications $\Phi = (P_{pre}, Q_{post})$, and prompts the LLM to generate an initial proof $\Pi_0$. For initial proof generation, we directly reuse the prompt of the first phase of AUTOVERUS (Yang et al., 2025b). EXVERUS then iteratively fixes proof errors via counterexample generation (Section 3.1) and mutation-based counterexample-guided repair (Section 3.2), until the proof passes the Verus verification, or until it reaches the max attempts.

## 3.1. Counterexample Generation with Validation

Given a target verification error $e_t$, EXVERUS first tries to synthesize a source-level SMT query (in Z3Py) $Q_t$ that produces multiple counterexamples $\Sigma_t$. Moreover, if $e_t$ is an error related to invariants, EXVERUS will invoke a validation module to filter out invalid counterexamples, enabling more grounded repair guided by validated counterexamples.

**Counterexample generation.** EXVERUS prompts the LLM with the buggy proof $\Pi_t$ and the invariant error $e_t$, instructing it to translate the Verus proof annotations into an SMT query (in Z3Py) $Q_t = QuerySyn(\text{LLM}, \Pi_t, e_t)$. Specifically, EXVERUS first constructs a comprehensive source-level SMT query generation prompt template. The prompt instructs the LLM to 1) faithfully translate the proof annotations into Z3Py constraints, 2) encode semantic information such as types in the naming convention (for reconstruction), 3) simplify constraints by only focusing on the failing assertion/invariant and the relevant proof annotations, 4) adaptively concretize some variables and avoid quantifiers, and 5) store the concrete variable assignment in a serializable list. The prompt can be found in Appendix J.1.

Note that counterexample generation is not guaranteed to succeed due to the LLM's inherent unreliability. Therefore, when EXVERUS fails to produce enough counterexamples, EXVERUS iteratively regenerates SMT queries by reflecting on the prior failures and query execution results to obtain a set of high-quality counterexamples $\Sigma_t = Solve(Q_t)$.

After obtaining enough SMT-generated counterexamples, EXVERUS optionally invokes the validation module to check whether they are truly counterexamples that reveal the verification failure (for invariant errors).

**Counterexample validation.** Due to the non-determinism of LLMs and potential threat of hallucination, the generated counterexamples are not guaranteed to be real counterexamples w.r.t. the verification errors. EXVERUS leverages a non-LLM verifier-based validation module to validate counterexamples for invariant errors due to the ease of task formulation, while leaving the validation for other types of errors as future work. That being said, the unchecked counterexamples can still serve as approximate, structured reasoning steps to guide the proof repair. We develop the validation module for invariant errors since invariant generation is a long-standing central challenge in verification, and is recognized as a major bottleneck by prior works (Flanagan & Leino, 2001; Garg et al., 2014; Kamath et al., 2023).

Specifically, the validation module consists of three steps:

1. Loop extraction: it isolates and extracts the loop body of the loop containing invariant into a standalone function, denoted as `loop_func`.
2. Invariant translation: it then translates the loop invariants into assertions both before the loop body and after the loop body, mimicking one loop execution with invariant checking. We denote the assertions as loop-start assertions and loop-end assertions, respectively.
3. Counterexample instrumentation: it instruments the function `loop_func` and injects the value assignments of a counterexample at the beginning of the function, e.g., Figure 3.

The counterexample-injected `loop_func` (denoted as `loop_func_injected`) is then checked by Verus, and any assertion error would be captured. Specifically, EXVERUS expects different symptoms for different invariant failures:

1. `InvFailFront` The invariant cannot be established at loop entry (i.e., it already fails before executing the loop body). For this error, EXVERUS expects a (reachable) counterexample that violates the corresponding loop-start assertion.
2. `InvFailEnd` The invariant holds at loop entry, but it is not preserved by one loop iteration, indicating the invariant is not inductive. For this error, EXVERUS expects a counterexample that passes the loop-start assertion, but fails the loop-end assertion.

EXVERUS captures any assertion errors and checks whether the corresponding symptoms are triggered. If so, the counterexample is considered validated. The validated counterexamples are passed to the mutation-based counterexample-guided repair module. In the following, we describe our recipe for the automated proof repair based on mutating existing proofs to block the generated counterexamples.

## 3.2. Mutation-based Counterexample-guided Repair

Given the set of distinct counterexamples, EXVERUS diagnoses the root cause of the proof failures and generates a repair. It (1) categorizes the failure via an LLM-based error triage module, (2) generates candidate repairs based on mutation with a specialized mutator $M_t \in M_{all}$, and (3) ranks the candidates using verifier feedback (and counterexample-validation feedback for invariant errors).

**Counterexample-based error triage.** EXVERUS queries an LLM with the buggy proof, the counterexamples, and verifier feedback to categorize the error. The triage analyzes whether the counterexamples are reachable from a valid initial state, i.e., suggesting the invariant/assertion is incorrect and should be replaced/relaxed, or are spurious, i.e., suggesting it should be strengthened. It outputs a verdict $v_t$ and a rationale $r_t$. Formally, $v_t, r_t = ErrorTriage(\text{LLM}, \Pi_t, e_t, \Sigma_t)$.

**Customized mutation.** Based on the triage $v_t$, EXVERUS selects a corresponding mutator, i.e., $M_t = MutatorSelect(M_{all}, v_t)$, and applies it to the buggy

proof. A strengthen-based mutator targets invariants that are correct but not inductive, as well as assertion failures (or post-condition violations) due to missing assertions. A replace-based mutator targets invariants or assertions that are factually wrong on reachable states. In both cases, the prompt provides few-shot repair patterns and includes the counterexamples and the triage rationale $r_t$ to encourage fixes that block the counterexamples. This produces a set of mutants $C_t = M_t(\text{LLM}, \Pi_t, e_t, \Sigma_t, r_t)$.

**Mutant ranking.** Inspired by the PDR algorithm (Bradley, 2011), EXVERUS uses multiple counterexamples to better characterize the failure and guide repair (see Section 6). For invariant errors, we score candidates by the number of validated counterexamples they block. A candidate is said to *block* a counterexample if the counterexample no longer triggers the corresponding invariant failure under the updated invariant. For non-invariant errors, EXVERUS falls back to the number of verified sub-goals (from Verus), similar to AUTOVERUS (Yang et al., 2025b). EXVERUS ranks candidates by this score and selects the best one for the next iteration. Formally, $\Pi_{t+1} = RankTop(C_t)$.

## 4. Experiment

### 4.1. Evaluation Setup

**Baselines.** We evaluate our approach against two baselines:

- **AutoVerus** (Yang et al., 2025b), the state-of-the-art LLM-based system for Verus proof generation. We use the same setting as presented in AUTOVERUS.
- **Iterative Refinement**, an iterative refinement method inspired by Shefer et al. (2025). In each iteration, the approach prompts the LLM with the unverified code, the corresponding error message from Verus, and a dedicated repair prompt (shown in Appendix J.3).

Other recent works (Aggarwal et al., 2025; Zhong et al., 2025; Chen et al., 2025) are not included because they have either different objectives and experimental setups, or did not publicly release their models and code.

**Metrics.** We use the success rate as the primary metric. We also include the number of input and output tokens to measure the cost.

**Dataset.** We curate benchmarks consisting of Verus proof tasks from the following sources:

- **VerusBench** (Yang et al., 2025b). A dataset contains proof tasks translated from different formal verification benchmarks such as MBPP-DFY-153, CloverBench, Diffy, and examples from the Verus documentation[2].

- **Dafny2Verus** (Aggarwal et al., 2025). This dataset consists of 67 tasks from the DafnyBench dataset (Loughridge et al., 2025) translated to Verus (see dataset filtering process detailed in Appendix G.2).
- **Leetcode-Verus** (Dai, 2025). This dataset comprises 28 challenging proof tasks derived from the LeetCode platform. The collection is curated by human experts who manually translate a set of LeetCode problems into Verus proofs. These complex tasks require extensive reasoning, with $\sim$200 LoC on average.
- **HumanEval-Verus** (Bai et al., 2025). This collection is part of an open-source effort to translate tasks from the HumanEval benchmark (Chen et al., 2021) to Verus. We curate the tasks using a similar approach to that described in AlphaVerus, resulting in 68 tasks.

**Models and parameters.** We use several state-of-the-art Large Language Models (LLMs), including Claude-Sonnet-4.5 (Anthropic, 2025), GPT-4o (OpenAI, 2024), o4-mini (OpenAI, 2025), Qwen3 Coder (`Qwen3-480B-A35B`) (Yang et al., 2025a), and DeepSeek-V3.1 (DeepSeek-AI, 2024). For all LLM inference tasks, we set the temperature to 1.0 following AUTOVERUS (Yang et al., 2025b) for a fair comparison. The maximum number of repair iterations is set to 10. The number of LLM responses in mutant generation in mutation-based counterexample-guided repair is set to 5.

**Implementation.** EXVERUS is implemented in Verus version `0.2025.07.12.0b6f3cb`. All experiments are conducted on a server running Ubuntu 22.04 LTS with an AMD EPYC 9554 CPU with 64 cores/128 threads and 1.1 TB RAM. Our implementation is based on Python ($\sim$13K LoC) and Rust ($\sim$2K LoC). For SMT solving, we use the Python Z3Py API (Bjørner et al., 2018) (version `4.15.1.0`). For counterexample validation, we develop parsing tools based on Rust Syn (version `v2.0.106`) and Verus Syn (version `v0.0.0-2025-08-12-1837`).

### 4.2. Main Results

**Overall performance.** Table 1 shows that EXVERUS consistently achieves leading performance across benchmarks and base models. On VerusBench, EXVERUS substantially outperforms AUTOVERUS[3] by 60.92% on average. On relatively easier benchmarks (VerusBench, DafnyBench), stronger LLMs (e.g., Sonnet-4.5) yield smaller gains over baselines than GPT-4o or DeepSeek-V3.1, suggesting that stronger intrinsic reasoning can partially compensate for counterexample reasoning. In contrast, on harder benchmarks the gap widens even with stronger models: EXVERUS

---

[2]Due to the rapid evolution of the Verus tool chain, four out of the original 150 tasks can no longer be verified, thus we end up with a total of 146 tasks.

[3]AUTOVERUS was evaluated on a now-deprecated version of Verus and thus suffers from performance degradation on the current Verus toolchain, as discussed in Appendix H.6.

*Table 1.* Repair success rate across different methods, models, and benchmarks. All rates are in percentages. Percentages in braces denote how EXVERUS improves over the best baseline among Iterative Refinement and AUTOVERUS.

| | | DeepSeek-V3.1 | GPT-4o | Qwen3-Coder | o4-mini | Sonnet-4.5 |
|---|---|---|---|---|---|---|
| **VerusBench** | Iterative Refinement | 60.3 | 43.2 | 69.2 | 69.2 | 83.6 |
| | AUTOVERUS | 24.7 | 39.0 | 51.4 | 32.2 | 75.3 |
| | EXVERUS | **71.9** (↑ **19.3%**) | **51.4** (↑ **19.0%**) | **71.9** (↑ **4.0%**) | **74.7** (↑ **7.9%**) | **88.4** (↑ **5.7%**) |
| **DafnyBench** | Iterative Refinement | 73.1 | 82.1 | 89.6 | 82.1 | **95.5** |
| | AUTOVERUS | 76.1 | 79.1 | 86.6 | 77.6 | **95.5** |
| | EXVERUS | **88.1** (↑ **15.7%**) | **88.1** (↑ **7.3%**) | **95.5** (↑ **6.7%**) | **95.5** (↑ **16.4%**) | **95.5** |
| **LCBench** | Iterative Refinement | **10.7** | **10.7** | 7.1 | 14.3 | 25.0 |
| | AUTOVERUS | **10.7** | 7.1 | **10.7** | 10.7 | 14.3 |
| | EXVERUS | **10.7** | **10.7** | **10.7** | **25.0** (↑ **75.0%**) | **28.6** (↑ **14.3%**) |
| **HumanEval** | Iterative Refinement | 11.8 | 8.8 | 19.1 | 20.6 | 29.4 |
| | AUTOVERUS | 14.7 | **14.7** | 16.2 | 20.6 | 27.9 |
| | EXVERUS | **17.6** (↑ **20.0%**) | **14.7** | **22.1** (↑ **15.4%**) | **30.9** (↑ **50.0%**) | **41.2** (↑ **40.0%**) |

solves about 2× and 1.5× as many tasks as AUTOVERUS on LCBench and HumanEval, respectively. We further analyze the overlap and complementarity between EXVERUS and AUTOVERUS in Appendix H.

**Robustness.** To address concerns about LLM memorization, we evaluate EXVERUS under code obfuscation. We build ObfsBench by obfuscating samples (both programs and proofs) from VerusBench (see Appendix E), generating 266 challenging yet verifiable out-of-distribution tasks.

As shown in Table 2, EXVERUS consistently outperforms AUTOVERUS across all ObfsBench subsets and various model configurations. Across the evaluated models (except GPT-4o), EXVERUS remains robust to all obfuscation strategies, achieving success rates above 73%, whereas AUTOVERUS remains below 40%. These results suggest that AUTOVERUS's heuristics-heavy prompting is less robust to out-of-distribution tasks, whereas EXVERUS better preserves semantic reasoning under code transformations.

**Cost.** Cost analysis shows that the average input and output tokens of EXVERUS are 113.4K and 16.2K respectively, which is 32.14% and 18.19% less than AUTOVERUS's 167.1K and 19.8K tokens. In total, EXVERUS costs 30.66% less tokens than AUTOVERUS (both with DeepSeek-V3.1 on VerusBench). We leave more fine-grained cost analysis in Appendix I.

**Ablations.** To investigate the effects of the error-specific mutators and the validation module, we designed a baseline that instructs the LLM to directly fix the proof based on the counterexamples without validation, denoted as EXVERUS_NO_MUT. To make this baseline competitive, we encode expert knowledge on how to repair different proof errors comprehensively into the prompt (see Appendix J.6). We also include Iterative Refinement as a reference.

Table 3 shows the importance of the counterexample-guided

mutation and validation in EXVERUS. The full EXVERUS pipeline outperforms EXVERUS_NO_MUT across nearly all scenarios. On VerusBench, the full system boosts the pass rate from 64.4% to 71.9% with DeepSeek-V3.1. This performance gap is even more significant on the robustness benchmark ObfsBench. The counterexample-guided mutation and validation module increases the pass rate from 65.4% to 81.6%. We perform fine-grained case analysis on two successful cases in Appendix A to demonstrate how each module in EXVERUS works.

We also observe that the ablated variant EXVERUS_NO_MUT is not uniformly stronger than Iterative Refinement: it underperforms Iterative Refinement 8 out of 25 model-benchmark combinations. We attribute this performance drop compared with EXVERUS to the mutation module and validation module. The mutation module uses the triaged error type to select a targeted action, and the validation module then filters candidates by checking whether they block the generated counterexamples. Without these modules, EXVERUS_NO_MUT relies solely on the LLM to interpret the counterexamples and figure out how to fix the proof, which is a much more open-ended and error-prone task.

### 4.3. Sensitivity Analysis

**Impact of number of counterexamples.** We study the effect of counterexamples via a controlled single-repair experiment focusing on invariant errors. Specifically, we curate InvariantInjectBench: 187 near-correct buggy proofs, each fixable by changing exactly one invariant (details in Appendix G).[4] We run both EXVERUS (using 10 counterexamples by default) and a variant of EXVERUS that uses one counterexample, denoted as EXVERUS_ONE_CEX, with one repair attempt. Out of 187 tasks, EXVERUS proves

---

[4]We also attempted to extract intermediate proofs from AutoVerus trajectories, but found few usable cases.

*Table 2.* Performance on all obfuscated programs (ExVERUS / AutoVERUS). All results are success rates in percentages.

| Category | Sub-strategy | All Obfuscated Programs | | | | |
|---|---|---|---|---|---|---|
| | | DeepSeek-V3.1 | GPT-4o | o4-mini | Qwen3-Coder | Sonnet-4.5 |
| Layout | Identifier Renaming | 81.5 / 25.9 | 50.0 / 31.5 | 74.1 / 25.9 | 81.5 / 38.9 | **87.0 / 66.7** |
| Data | Dead Variables | 81.7 / 27.9 | 40.8 / 20.4 | 79.2 / 17.1 | 76.2 / 30.4 | **90.4 / 62.9** |
| | Instruction Substitution | 79.9 / 24.7 | 42.9 / 24.0 | 78.6 / 18.8 | 73.4 / 31.8 | **90.3 / 66.2** |
| Control Flow | Dead Code Insertion | 73.9 / 30.4 | 26.1 / 8.7 | **87.0** / 8.7 | 65.2 / 21.7 | 78.3 / **56.5** |
| | Opaque Predicates | 86.4 / 31.8 | 27.3 / 18.2 | 86.4 / 13.6 | 77.3 / 31.8 | **90.9 / 77.3** |
| | Control Flow Flattening | 86.5 / 36.5 | 28.8 / 21.2 | 80.8 / 11.5 | 78.8 / 17.3 | **92.3 / 69.2** |

*Table 3.* Ablation study on mutation strategies. The results are success rates in percentage. Percentages in braces denote how ExVERUS improves over the best baseline results among Iterative Refinement and AutoVERUS.

| | | DeepSeek-V3.1 | GPT-4o | Qwen3-Coder | o4-mini | Sonnet-4.5 |
|---|---|---|---|---|---|---|
| **VerusBench** | Iterative Refinement | 60.3 | 43.2 | 69.2 | 69.2 | 83.6 |
| | ExVERUS_NO_MUT | 64.4 | 46.6 | 65.8 | 68.5 | 84.9 |
| | **ExVERUS** | **71.9 (↑ 11.7%)** | **51.4 (↑ 10.3%)** | **71.9 (↑ 4.0%)** | **74.7 (↑ 7.9%)** | **88.4 (↑ 4.0%)** |
| **DafnyBench** | Iterative Refinement | 73.1 | 82.1 | 82.1 | 89.6 | **95.5** |
| | ExVERUS_NO_MUT | **88.1** | **89.6** | 92.5 | 85.1 | **95.5** |
| | **ExVERUS** | **88.1** | 88.1 | **95.5 (↑ 3.2%)** | **95.5 (↑ 6.7%)** | **95.5** |
| **LCBench** | Iterative Refinement | **10.7** | **10.7** | **14.3** | 7.1 | 25.0 |
| | ExVERUS_NO_MUT | 7.1 | **10.7** | 10.7 | 17.9 | 21.4 |
| | **ExVERUS** | **10.7** | **10.7** | 10.7 | **25.0 (↑ 40.0%)** | **28.6 (↑ 14.3%)** |
| **HumanEval** | Iterative Refinement | 11.8 | 8.8 | 20.6 | 19.1 | 29.4 |
| | ExVERUS_NO_MUT | **17.6** | 8.8 | 19.1 | 22.1 | 29.4 |
| | **ExVERUS** | **17.6** | **14.7 (↑ 66.7%)** | **22.1 (↑ 7.1%)** | **30.9 (↑ 40.0%)** | **41.2 (↑ 40.0%)** |
| **ObfsBench** | Iterative Refinement | 61.3 | 28.6 | 71.4 | 69.9 | 86.8 |
| | ExVERUS_NO_MUT | 65.4 | 35.3 | 71.8 | 72.9 | 85.7 |
| | **ExVERUS** | **81.6 (↑ 24.7%)** | **41.0 (↑ 16.0%)** | **76.7 (↑ 6.8%)** | **79.7 (↑ 9.3%)** | **90.6 (↑ 4.3%)** |

106 tasks while ExVERUS_ONE_CEX proves 100, showing that more counterexamples are contributing positively to counterexample-guided repair.

**Discriminative power of validation module.** To evaluate validation via counterexample-blocking, we count blocked counterexamples per mutant and track verification and task repair. On InvariantInjectBench, blocking counterexamples strongly correlates with success. For ExVERUS, mutants blocking 0 counterexamples pass verification in 32/83 (38.55%) and repair 9/21 tasks (42.86%), whereas mutants blocking ≥ 1 counterexample verify in 158/245 (64.49%) and repair 41/51 tasks (64.49%). For ExVERUS_ONE_CEX, blocking 0 counterexamples yields 38/153 (24.84%) verified and 12/36 tasks (33.33%) repaired, while blocking the (single) counterexample yields 172/243 (70.78%) verified and 45/53 tasks (84.91%) repaired. Overall, counterexample-blocking effectively filters good mutants, demonstrating the discriminative power of ExVERUS's verification module.

## 5. Discussion and Limitations

**Counterexample validation beyond loop invariants.** Our prover-based counterexample validation targets specifically for invariants because invariant inference is recognized as one of the most prevalent bottlenecks for verification (Flanagan & Leino, 2001; Garg et al., 2014; Kamath et al., 2023). However, validating counterexamples for other errors, such as assertion errors, goes beyond our validation module's capabilities, as they are sometimes not well-defined, e.g., an assertion error could be caused by a missing `trigger` annotation. That said, the unvalidated counterexamples could still help LLMs propose repairs, where they fall back to more structured reasoning steps, so ExVERUS still demonstrates improved repair performance empirically for other errors guided by counterexamples.

**Scope beyond Verus.** ExVERUS is designed and evaluated for Verus, and we do not claim generality to every verifier. However, the challenge it addresses is prevalent. SMT-backed verifiers often translate source programs into low-level representations before solving, which cannot be reconstructed into actionable source-level feedback. Therefore,

the philosophy of generating and validating source-level counterexamples has the potential to benefit other verifiers.

**Initial proof generation.** We keep the initial proof generation stage simple by reusing AUTOVERUS's prompt for a fair comparison. Our focus is on the downstream counterexample-driven repair and generalization components; improving initial proof generation via prompt engineering (Yang et al., 2025b) or finetuning (Chen et al., 2025) is complementary to EXVERUS.

## 6. Related Work

**LLM for automated verification.** Recent LLM-based systems have shown superior performance in proof generation and repair for both interactive theorem proving, e.g., Rocq (Lu et al., 2024; Kozyrev et al., 2024), Lean (Li et al., 2026; Yang et al., 2023; Song et al., 2023; 2024), and whole proof generation, e.g., Isabelle (First et al., 2023), Verus (Yang et al., 2025b; Chen et al., 2025; Aggarwal et al., 2025), Dafny (Banerjee et al., 2026).

Existing techniques on Verus proof synthesis follow the paradigm of prompting the LLM to generate proof annotations and iterative repair verification failures based on verifier feedback (Zhong et al., 2025; Yang et al., 2025b; Yao et al., 2023; Aggarwal et al., 2025; Chen et al., 2025). Unfortunately, the verifier feedback is often too coarse and ambiguous to reveal the root cause of the verification failure. To better leverage verifier feedback, AUTOVERUS (Yang et al., 2025b) encodes repair strategies as prompts for each error type, but these manually crafted strategies require frequent updates to adapt to new proof errors. SAFE (Chen et al., 2025) embeds the repair capabilities via training with synthetic data, but it incurs a nontrivial data curation cost, e.g., a month of non-stop GPT-4o invocations and rejection sampling. EXVERUS complements these approaches with concrete, actionable feedback by generating source-level counterexamples as part of the reasoning steps during repair.

**Counterexample-guided proof synthesis.** Counterexamples have long served as a building block for incremental proof synthesis. Techniques like Counterexample-guided Abstraction Refinement (CEGAR) (Clarke et al., 2000) and Property Directed Reachability (PDR) (Bradley, 2011) iteratively refine proofs by blocking counterexamples provided by solvers. However, applying these ideas to software verification, such as in systems developed in Rust, is challenging because source-level constructs are lost during the compilation of low-level verification conditions. Moreover, existing algorithms that adopt fixed templates, e.g., dropping literals, often fail to generalize a concrete counterexample over infinite state space (e.g., integers, heaps, etc.) into a blocking predicate. EXVERUS leverages multiple counterexamples with LLM-based proof mutations to improve the general-

ization of failure patterns. This allows it to incrementally propose and prioritize repairs that naturally align with the LLM-based iterative repair paradigm.

## 7. Conclusion

We presented EXVERUS, an automated LLM-based Verus proof repair framework guided by counterexamples. Unlike prior LLM-based systems that rely on static code and coarse verifier feedback, EXVERUS actively synthesizes, validates, and blocks counterexamples to guide proof refinement. By grounding LLM reasoning in concrete program behaviors, EXVERUS transforms open-ended proof search into a more grounded process. Extensive experiments across multiple Verus benchmarks, including our newly introduced ObfsBench for robustness evaluation, demonstrate that EXVERUS substantially outperforms the baselines in success rates, robustness, and cost efficiency.

## Acknowledgements

We thank all reviewers, Chenyuan Yang and Chenyu Zhou for their constructive and insightful comments and feedback, which significantly improved this paper. This research is supported by NSF awards CNS-2313190, CCF-2119184, and NSERC Discovery Grant RGPIN-2025-06870. It is also supported in part by the OpenAI Research Access Program (OpenAI, 2025).

## Impact Statement

This paper advances ML-assisted formal verification by introducing EXVERUS, a counterexample-guided framework that grounds LLM-based proof repair in concrete, verifier-validated counterexamples and generalizes them into inductive invariants to improve robustness and efficiency for Verus proofs. In the longer term, such tooling can lower the barrier to adopting formal methods and help more developers apply verification to safety and security-critical Rust systems, potentially reducing defects and improving reliability. However, the assurance provided by automated proof repair remains bounded by the quality and completeness of the specifications being verified. Users may overinterpret generated annotations as evidence of full correctness, even when the verified properties are incomplete, too weak, or misaligned with the intended behavior; similarly, automated proof construction could lend undeserved credibility to codebases whose specifications are selective or misleading. We therefore recommend deploying these methods with transparent specification assumptions, human-in-the-loop review for high-stakes settings, and clear governance on where automated proof-repair pipelines are appropriate.

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

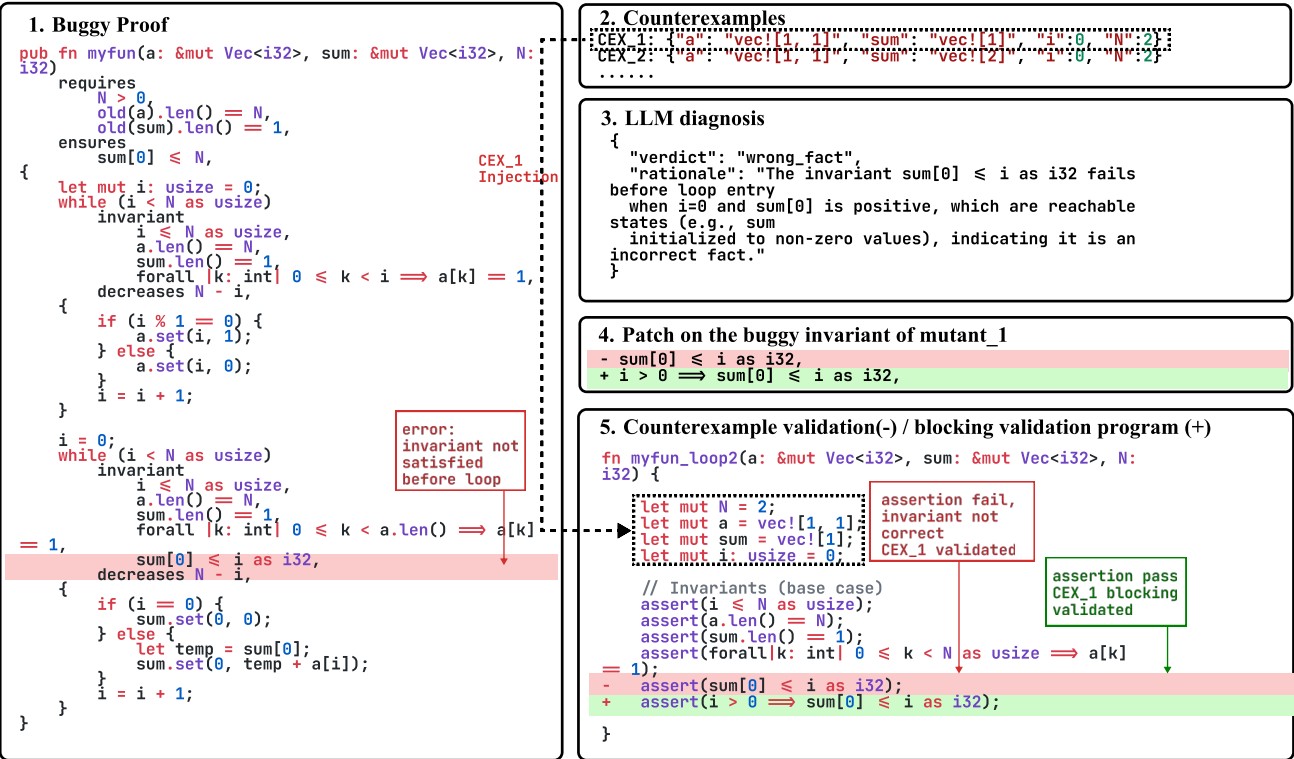

Figure 3. Repairing a wrong invariant that involves an invalid state by pinpointing and pruning it. Task Diffy/brs1 in VerusBench.

## A. Case Study

### A.1. Invariant Weakening via State Pruning

Figure 3 shows an almost-correct proof from VerusBench. Verus provides feedback "error: invariant not satisfied before loop" for the buggy invariant `sum[0] <= i`. This failure occurs because the LLM overlooked an edge case, i.e., in the first iteration, `sum` hasn't been initialized yet, so it can be any value. In every iteration after that, `sum[0] <= i` holds. The LLM realized that something like `sum[0] <= i` is necessary to prove the post-condition.

Although it appeared to be easy to solve, the state-of-the-art LLM-based proof generation tool, AUTOVERUS, failed to prove this task after 15 preliminary proof generation attempts (Phase 1), 4 generic proof refinement attempts (Phase 2), and 21 error-driven proof debugging attempts (Phase 3). After inspecting the trajectory of AUTOVERUS, we observed that AUTOVERUS spent 16 attempts (Phase 3) to fix "error: invariant not satisfied before loop", but none of them worked. This invariant error and the struggling repair process boil down to the fundamental limitation of lacking concrete, actionable feedback like counterexamples (Dougrez-Lewis et al., 2025; Cheng et al., 2024; Shojaee et al., 2025).

EXVERUS first synthesizes a Z3Py script to produce counterexamples. The error triage LLM figures out the counterexamples are reachable, meaning the invariant is "Incorrect" and needs a replacing mutator. In the mutation-based proof repair stage, it identified the pattern shared by the counterexamples: `i=0` and `sum[0]` is positive, and invoked the mutator to generate mutants that block this pattern. Finally, mutant-1 successfully blocks all counterexamples and passes Verus verification, resolving this task.

### A.2. Wrong Invariant Detection and Removal

Figure 4 shows another almost-correct proof from ObfsBench. AUTOVERUS failed to prove this task after 15 preliminary proof generation attempts (Phase 1), one generic proof refinement attempt (Phase 2), and 24 error-driven proof debugging attempts (Phase 3). AUTOVERUS spent 14 attempts (Phase 3) to fix assertion failures, but none of them worked.

The buggy invariant reports "error: invariant not satisfied before loop", indicating the invariant is incorrect. All counterexamples trigger the red assertion (translated from the buggy invariant) and are validated. The error triage LLM then reasons

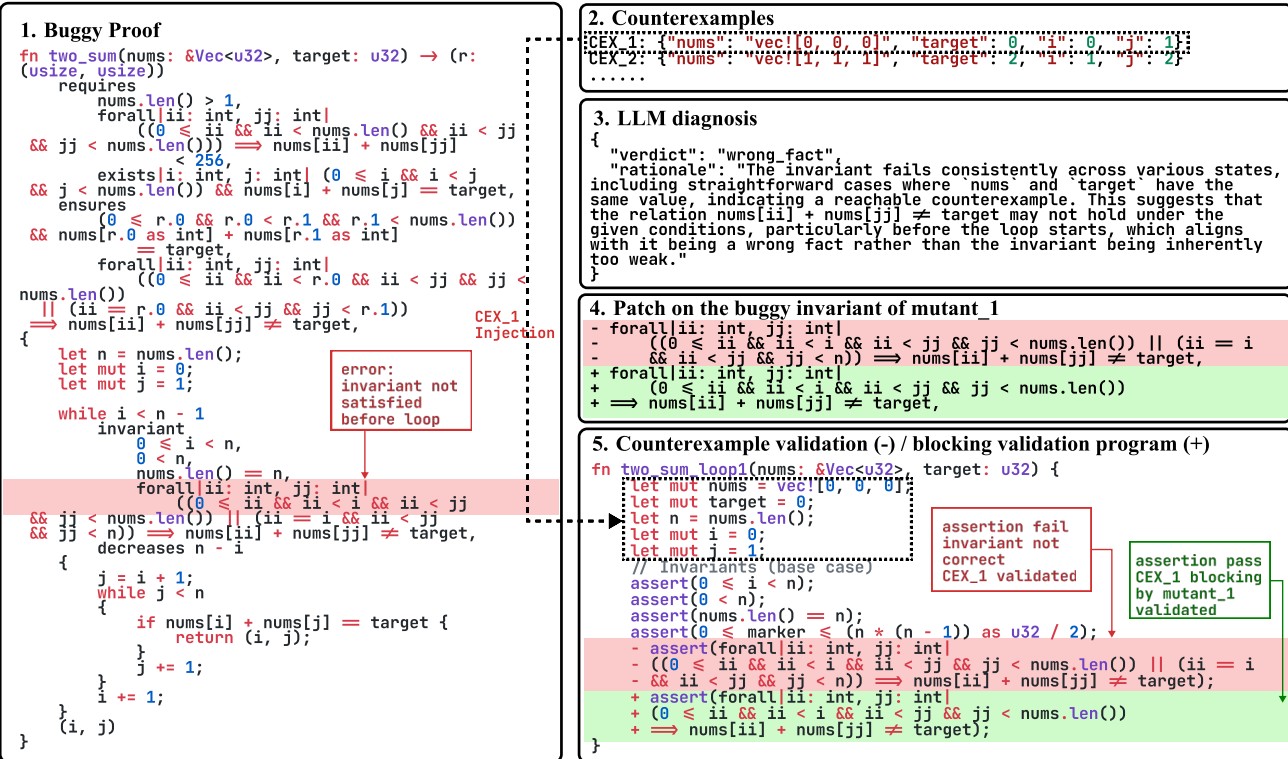

*Figure 4.* Identifying and removing a wrong invariant guided by counterexamples. Task CloverBench_two_sum_3 in ObfsBench.

about the validated counterexamples, summarizing that the counterexamples are reachable and labelling the invariant as "incorrect". Consequently, it invokes the replacing mutator and produces a set of mutants. The mutant_1 successfully blocks all counterexamples, i.e., it passes the green assertion, and passes Verus verification, solving the task.

To conclude, compared with coarse verifier messages, a counterexample provides concrete feedback by exhibiting a specific program state in which an invariant/assertion does not hold, immediately revealing the root cause, e.g., an overlooked edge case or a fundamentally wrong invariant. Guided by multiple counterexamples, EXVERUS converts debugging into a targeted search: candidate fixes that block them are prioritized, enabling incremental, step-by-step refinement that converges to the correct proof.

## A.3. A Challenging Case in VeruSAGE-Bench

To investigate how EXVERUS' counterexample reasoning could assist system-level proofs, we adapt the idea of EXVERUS into the repo-level verification with an agentic scaffold. To this end, we consider VeruSage (Yang et al., 2025c), a comprehensive Verus system verification benchmark suite with 800+ proof tasks extracted from eight open-source Verus-verified system projects, such as operating systems, distributed systems etc. Every task corresponds to one proof function or executable Rust function in the original project, with all the dependencies extracted into a stand-alone Rust file that can be individually compiled and verified. VeruSAGE-Bench is extremely complex and challenging, containing 947 LoC per task on average. Surprisingly, the best evaluated LLM-agent combination, i.e., using a generic coding agent (GitHub Copilot Command-Line Interface (GitHub, 2025)) and a simple prompt (Hands-Off Approach[5], shown in Appendix J.8.1) powered by Sonnet 4.5, successfully proved 81% of tasks. Despite demonstrating strong capability in system proof generation, several bottlenecks remained. For instance, Yang et al. (2025c) found that when Sonnet 4.5 failed to complete an Anvil Controller (Sun et al., 2024b) proof, the corresponding human-written proof uses an inductive invariant, indicating the inductive invariant generation capability is a bottleneck for Sonnet 4.5.

We extend Hands-Off Approach by rerunning Hands-Off Approach with a counterexample-enhanced prompt on the last failed attempt of Hands-Off Approach, denoted as **Counterexample-Augmented Hands-Off Approach**. Specifically,

---

[5]The prompt of Hands-Off Approach can be found in Yang et al. (2025c).

the counterexample-enhanced prompt (Appendix J.8.2) instructs the agent to reason about the current verification failure, generate a counterexample in both natural language and concrete value assignments, and understand the root cause of failure based on the counterexample. For comparison, we design an ablated version which reruns Hands-Off Approach with the original prompt on the last failed attempt of Hands-Off Approach, denoted as **Double Hands-Off Approach**. Below is a case where Hands-Off Approach failed, Double Hands-Off Approach also failed, but Counterexample-Augmented Hands-Off Approach succeeded. This task requires proving a temporal stability property about a Kubernetes VReplicaSet controller: once a property $q$ ("no deletion timestamp on VRS in `ongoing_reconciles`") holds, it persists forever, written as $spec \models \top \rightsquigarrow \Box q$ in TLA$^+$-style temporal logic (Lamport, 1994). The proof file contains 4,179 lines of Verus code with six available temporal-logic axiom lemmas. The key axiom, `leads_to_stable`, requires three preconditions:

  (i) **Stability**: $spec \models \Box (q \land next \Rightarrow q')$, i.e., the property is preserved by every transition,
 (ii) **Fairness**: $spec \models \Box next$, i.e., transitions always occur,
(iii) **Reachability**: $spec \models p \rightsquigarrow q$, i.e., the property is eventually reached.

Preconditions (ii) and (iii) follow readily from the lemma's `requires` clause, but Precondition (i) demands *lifting* a state-level argument to temporal-level reasoning, which is the central challenge of this proof.

**Where Hands-Off Approach got stuck.** In Step 1 (first attempt of Hands-Off Approach), the agent correctly identifies the high-level proof strategy: use `leads_to_weaken` to establish reachability, then `leads_to_stable` to convert it into persistence. However, it leaves the stability assertion's proof body empty (by `{}`), hoping the SMT solver will discharge it automatically (Listing 1). Verus rejects the proof because the empty body does not establish Precondition (i) of `leads_to_stable` (Listing 2).

*Listing 1.* Hands-Off Approach (Step 1): proof attempt. `q` denotes the target property and `inv` the schedule-level invariant.

```
 1 let p = |s| !s.ongoing_reconciles(cid)
 2              .contains_key(key);
 3 let q = vrs_ongoing_no_del_ts(vrs, cid);
 4 let inv = vrs_sched_no_del_ts(vrs, cid);
 5
 6 assert forall |s|
 7   #[trigger] p(s) implies q(s)
 8 by {}
 9
10 assert forall |s, s_prime|
11   q(s) && inv(s) && inv(s_prime)
12   && #[trigger] cluster.next()(s, s_prime)
13   implies q(s_prime)
14 by {}   // <-- EMPTY: stability unproven
15 leads_to_weaken(spec, true_pred(),
16   lift_state(p), true_pred(), lift_state(q));
17 leads_to_stable(spec,              // <-- ERROR
18   lift_action(cluster.next()),
19   true_pred(), lift_state(q));
```

*Listing 2.* Verus error for Listing 1.

```
 1 error: precondition not satisfied
 2   spec.entails(
 3     always(q.and(next).implies(later(q)))),
 4   --- failed precondition
 5   leads_to_stable(spec, ...);
 6   ^^^
```

Even if the state-level assertion were proven, Verus cannot automatically lift it to the temporal-level entailment required by `leads_to_stable`. The agent spends over 16 minutes exploring alternatives but ultimately concludes: *"this proof cannot be completed with the given set of axioms."*

In Double Hands-Off Approach (Step 2), given the failed output and error messages, the agent retries with two different strategies that also fail (Listing 3). In the first attempt, the agent decomposes the proof into three helper lemmas (for reachability, stability, and transitivity) but leaves all three with empty bodies, causing three "postcondition not satisfied" errors. In the second attempt, the agent calls the axiom lemmas directly but with incorrect arguments (e.g., passing the same

predicate as both source and target of `leads_to_stable`), causing two "precondition not satisfied" errors.

*Listing 3.* Double Hands-Off Approach (Step 2): two failed attempts.

```
1  // == Attempt 1: decompose into helper lemmas ==
2  lemma_vrs_ongoing_is_stable(spec, cluster, vrs, cid);
3  lemma_pre_implies_post(spec, vrs, cid);
4  leads_to_stable(spec,
5    lift_action(cluster.next()),
6    lift_state(pre), lift_state(post));
7  lemma_leads_to_trans_for_always(spec, vrs, cid);
8  // All 3 helper lemmas have EMPTY proof bodies
9  // => error: postcondition not satisfied (x3)
10
11 // == Attempt 2: wrong argument structure ==
12 leads_to_weaken(spec,
13   lift_state(not_in_ongoing),
14   lift_state(not_in_ongoing),   // <-- wrong
15   true_pred(), lift_state(state_pred));
16 leads_to_stable(spec,
17   lift_action(cluster.next()),
18   lift_state(state_pred),
19   lift_state(state_pred));      // <-- self-loop
20 // => error: precondition not satisfied (x2)
```

The common failure pattern across all attempts without counterexample guidance is that the agent cannot bridge the gap between *state-level* reasoning (`forall |s, s'| ...`) and the *temporal-level* entailment that `leads_to_stable` requires ($spec \models \square(\ldots)$). Without guidance, the agent either leaves this gap unfilled (empty bodies) or misuses the axiom API.

**How counterexamples guided the fix.** In Counterexample-Augmented Hands-Off Approach, the Step 2 prompt instructs the agent to generate a counterexample for the verification failure and use it to identify the root cause. The agent produces the counterexample in two formats.

First, in **concrete value assignments** (Listing 4), the agent instantiates the quantified variables with specific values, pinpointing the failing obligation: given a state $s$ where $q$ holds and a successor $s'$ via `cluster.next()`, we must prove $q(s')$, specifically that `deletion_timestamp` remains `None` in $s'$.

*Listing 4.* counterexample: concrete value assignments.

```
1  vrs = VReplicaSetView { object_ref: "vrs-123" }
2  controller_id = 0
3  s = ClusterState where:
4    s.ongoing_reconciles(0)["vrs-123"]
5      .triggering_cr.metadata
6      .deletion_timestamp = None  // q(s) = true
7  s_prime = ClusterState where
8    cluster.next()(s, s_prime) = true
9
10 MUST PROVE q(s_prime):
11   s_prime.ongoing_reconciles(0)["vrs-123"]
12     .triggering_cr.metadata
13     .deletion_timestamp is None
14 // Cannot be proven: empty proof body provides
15 // no reasoning about how next() affects
16 // ongoing_reconciles
```

Second, in a **natural-language explanation** (Listing A.3), the agent identifies the three pieces of reasoning that the empty proof body fails to provide: the relationship between the invariant `inv` (scheduled reconciles have no deletion timestamp), the target property $q$ (ongoing reconciles have no deletion timestamp), and the transition semantics of `cluster.next()`.

**counterexample: natural-language explanation (abbreviated).**

```
The proof needs to show that once q holds (the VRS in ongoing_reconciles has no deletion timestamp), it remains stable across
↪  all next transitions. The assertion tries to prove this but has an empty body (by {}), meaning Verus cannot derive the
↪  stability property.
```

```
The root cause is that the proof doesn't establish the connection between:
1. The invariant inv: scheduled_reconciles has no deletion timestamp
2. The property q: ongoing_reconciles has no deletion timestamp
3. How cluster.next() preserves property q

MISSING PROOF STEPS:
- When ongoing_reconciles changes via run_scheduled_reconcile, triggering_cr comes from scheduled_reconciles (satisfies inv)
- The triggering_cr field is immutable once set (only updated via continue_reconcile, which preserves it)
```

With the counterexample pinpointing the exact gap, the agent follows a systematic reasoning chain below (Listing A.3) that leads to the fix. The key insight, directly prompted by the counterexample's identification of the missing "connection between `inv` and $q$," is that the agent needs a *helper lemma* to extract $\text{always}(inv).\texttt{satisfied\_by}(ex)$ from the precondition $spec.\texttt{entails}(\text{always}(inv))$ for a specific execution trace.

---

**Agent's reasoning chain in Counterexample-Augmented Hands-Off Approach (condensed).**

```
1. COUNTEREXAMPLE ANALYSIS:
   State-level assertion (forall |s, s'| ...)
   does not establish temporal entailment
   (spec |= always(...)). This is the root cause.

2. STATE-LEVEL STABILITY (case analysis):
   Case 1: VRS already in ongoing_reconciles(s)
     -> triggering_cr preserved
     -> q(s) implies q(s')
   Case 2: VRS added via run_scheduled_reconcile
     -> triggering_cr := scheduled_reconciles[key]
     -> inv(s) implies q(s')

3. TEMPORAL BRIDGE (the missing piece):
   Need: always(inv).satisfied_by(ex)
   Have: spec.entails(always(inv))
   Gap:  no existing axiom connects these
   -> Create helper lemma to unfold entails:
       spec.entails(always(inv))
        /\ spec.satisfied_by(ex)
        ==> always(inv).satisfied_by(ex)

4. TWO-LAYER PROOF STRUCTURE:
   Layer 1: Prove always(q /\ inv /\ next
             => later(q))
   Layer 2: Since inv always holds, drop inv
         to get always(q /\ next => later(q))
```

---

The resulting fix (Listing 5) introduces a small helper lemma that bridges the `entails`/`satisfied_by` gap, then uses it in a two-layer temporal proof to establish Precondition (i).

This case illustrates two ways counterexample reasoning helps. *First*, it forces the agent to **concretize the failure**: by writing down specific variable values and tracing the failing obligation, the agent identifies the precise semantic gap (state-level vs. temporal-level reasoning). *Second*, it provides **actionable repair guidance**: the counterexample's identification of "missing proof steps" (how `inv` relates to $q$ through `run_scheduled_reconcile`) directly helps the agent construct the case analysis and the helper lemma that bridges the gap. Notably, the agent's solution differs from the human-written ground truth, which uses a `combine_spec_entails_always_n!` macro to fold invariants into a strengthened transition relation. The agent instead derives the same result from first principles via the helper lemma, a valid but structurally different proof, demonstrating that counterexample-guided reasoning leads to correct solutions rather than merely imitating reference proofs. This also suggests that counterexamples can help not only with loop inductive invariants, but also with the more challenging task of generating and repairing *temporal* invariants in system-level verification.

*Listing 5.* The fix produced by Counterexample-Augmented Hands-Off Approach. Top: new helper lemma. Bottom: key excerpt of the two-layer temporal proof.

```
 1  // == Helper lemma (new, 8 lines) ==
 2  proof fn lemma_from_entails_always_helper<T>(
 3    spec: TempPred<T>,
 4    inv: TempPred<T>,
 5    ex: Execution<T>)
 6    requires spec.entails(always(inv)),
 7             spec.satisfied_by(ex),
 8    ensures  always(inv).satisfied_by(ex),
 9  {
10    assert((spec.implies(always(inv)))
11      .satisfied_by(ex));
12  }
13
14  // == Main proof body (key excerpt) ==
15  // Layer 1: stability with inv explicit
16  assert forall |ex| spec.satisfied_by(ex)
17    implies always(
18      q.and(inv).and(next).implies(later(q)))
19      .satisfied_by(ex)
20  by {
21    lemma_from_entails_always_helper(
22      spec, lift_state(inv), ex);
23    // state-level case analysis now proven
24    ...
25  }
26  // Layer 2: drop inv (it always holds)
27  assert forall |ex| spec.satisfied_by(ex)
28    implies always(
29      q.and(next).implies(later(q)))
30      .satisfied_by(ex)
31  by {
32    lemma_from_entails_always_helper(
33      spec, lift_state(inv), ex);
34    // inv at every suffix -> redundant
35    ...
36  }
37  leads_to_stable(spec,
38    lift_action(cluster.next()),
39    true_pred(), lift_state(q));
40  // => verification results: 2 verified, 0 errors
```

## B. Pseudo-Code of EXVERUS

---

**Algorithm 1** EXVERUS Pipeline

---

1: **procedure** EXVERUS($\mathcal{P}, \Phi, model, MaxAttempts, \textsc{MaxZ3}, k$)
2:     $\Pi_0 \leftarrow \textsc{InitProofGen}(\mathcal{P}, \Phi, model)$
3:     $(st, \ell) \leftarrow \textsc{Verify}(\mathcal{P}, \Phi, \Pi_0)$
4:     **if** $st = \textsc{Pass}$ **then**
5:         **return** {$\Pi_0$, status=PASS, phase=init_gen}
6:     **end if**
7:     $\Pi \leftarrow \Pi_0$
8:     **for** $t \leftarrow 1$ to *MaxAttempts* **do**
9:         $(st, \ell) \leftarrow \textsc{Verify}(\mathcal{P}, \Phi, \Pi)$    ▷ $st, \ell$ refer to status and verification log
10:         **if** $st = \textsc{Pass}$ **then**
11:             **return** {$\Pi$, status=PASS, phase=cex_repair}
12:         **end if**
13:         **if** $st = \textsc{CompileError}$ **then**
14:             $\Pi \leftarrow \textsc{CompilationFixer}(\Pi, \ell, model)$
15:             **continue**
16:         **end if**
17:         $e_t \leftarrow \textsc{ExtractAndPrioritizeErr}(\ell)$
18:         $\Sigma_t \leftarrow \textsc{CexGen}(\Pi, e_t, model, k, \textsc{MaxZ3})$
19:         **if** $\textsc{IsInvariantErr}(e_t) \wedge \Sigma_t \neq \varnothing$ **then** ▷ Check if $e_t$ is an invariant bug and $\Sigma_t$ is not empty
20:             $\Sigma_t^{val} \leftarrow \textsc{ValidateCex}(\mathcal{P}, \Phi, \Pi, e_t, \Sigma_t)$
21:         **else**
22:             $\Sigma_t^{val} \leftarrow \Sigma_t$
23:         **end if**
24:         $\Pi' \leftarrow \textsc{MutValRepair}(\Pi, e_t, \Sigma_t^{val}, model)$
25:         **if** $\Pi' = \varnothing$ **then**
26:             **continue**
27:         **else**
28:             $\Pi \leftarrow \Pi'$
29:         **end if**
30:     **end for**
31:     $(st, \ell) \leftarrow \textsc{Verify}(\mathcal{P}, \Phi, \Pi)$
32:     **if** $st = \textsc{Pass}$ **then**
33:         **return** {$\Pi$, status=PASS, phase=cex_repair}
34:     **else**
35:         **return** {$\Pi$, status=FAIL, phase=cex_repair}
36:     **end if**
37: **end procedure**

---

**Algorithm 2** Counterexample Generation ($\Sigma_t = \textsc{Solve}(z3py_t)$)

---

1: **procedure** CEXGEN($\Pi_t, e_t, model, k, \textsc{MaxZ3}$)
2:     **for** $i \leftarrow 1$ to $\textsc{MaxZ3}$ **do**
3:         $Q_t \leftarrow \textsc{MakeCexPrompt}(\Pi_t, e_t, k)$    ▷ $Q_t$ is a query-generation *prompt*
4:         $z3py_t \leftarrow \textsc{QuerySyn}(Q_t, model)$   ▷ LLM translates $Q_t$ to a Z3Py script
5:         $(status, raw) \leftarrow \textsc{RunZ3}(z3py_t)$
6:         **if** $status \neq \textsc{Sat}$ **then**
7:             $Q_t \leftarrow \textsc{Feedback}(Q_t, status)$
8:             **continue**
9:         **end if**
10:         $norm \leftarrow \textsc{Normalize}(raw)$    ▷ normalize format
11:         **if** $\neg\textsc{SemanticValid}(norm, \Pi_t)$ **or** $|norm| < k/2$ **then**
12:             $Q_t \leftarrow \textsc{Feedback}(Q_t, \textsc{gatefail})$
13:             **continue**
14:         **end if**
15:         **return** $\textsc{MakeCex}(norm, e_t)$    ▷ returns $\Sigma_t$
16:     **end for**
17:     **return** $\varnothing$
18: **end procedure**

---

**Algorithm 3** Mutation-based Counterexample-guided Repair

---

1: **procedure** MUTVALREPAIR($\Pi_t, e_t, \Sigma_t^{val}, model$)
2:     $(v_t, r_t) \leftarrow \textsc{ErrorTriage}(\Pi_t, e_t, \Sigma_t^{val}, model)$
3:     $M_t \leftarrow \textsc{MutatorSelect}(M_{all}, v_t)$
4:     $\mathcal{C}_t \leftarrow \textsc{ApplyMutator}(M_t, \Pi_t, e_t, \Sigma_t^{val}, r_t, model)$
5:     **if** $\mathcal{C}_t = \varnothing$ **then**
6:         **return** $\varnothing$
7:     **end if**
8:     $\mathcal{C}_t \leftarrow \textsc{FilterCompilable}(\mathcal{C}_t)$
9:     **if** $\textsc{AnyPass}(\mathcal{C}_t)$ **then**
10:         **return** $\textsc{FirstPass}(\mathcal{C}_t)$
11:     **end if**
12:     **return** $\textsc{RankTop}(\mathcal{C}_t, \Sigma_t^{val}, e_t)$
13: **end procedure**

---

*Figure 5.* Pseudo-code of EXVERUS. Algorithm 1 illustrates the overall pipeline, Algorithm 2 illustrates counterexample generation, and Algorithm 3 illustrates mutation-based counterexample-guided repair.

## C. Software and Data

An anonymized artifact accompanying this paper is available at https://github.com/claudeyj/exverus. The repository contains all datasets and the complete implementation of the EXVERUS pipeline used in our experiments, including scripts for counterexample generation, validation, and evaluation. The datasets cover VerusBench, DafnyBench, LCBench, HumanEval, and our robustness benchmark ObfsBench.

This artifact will be submitted for *Artifact Evaluation*. While the pipeline code and datasets are fixed, reproducing end-to-end results requires running large language model (LLM) inference. Consequently, re-runs may incur token costs and exhibit small variations in quantitative metrics (e.g., success rate, token usage) due to the stochasticity of LLM generation and provider-side updates. We provide scripts and configuration files to replicate our evaluation protocol. However, exact numerical values may not match the paper's numbers bit for bit. Qualitative findings and comparative trends are expected to remain consistent.

# D. Initial Proof Generation Setting

We initiate our pipeline with a preliminary proof generation step, as shown from line 2 to line 4 in Algorithm 1. For initial proof generation, we directly reuse the prompt of the initial proof generation phase of AUTOVERUS (Yang et al., 2025b) as it is the state-of-the-art LLM-based proof generation tool (implementation details can be found in Appendix D). This initial proof synthesis is conducted using a straightforward LLM generation strategy. We employ the same prompt as the one used in the *preliminary proof generation* phase of AutoVerus (Yang et al., 2025b) for easier and fair comparison. If the initial generation does not pass the verification, it proceeds into the iterative repair process, i.e., Module 2 and 3, until the proof is repaired or the maximum attempts are reached (10 in our paper). If a proof in the iterations falls into compilation errors, e.g., syntax errors or type mismatch, the prompting-based compilation fixer will be invoked in the next iteration, to deliberately fix the compilation error, since Modules 2 and 3 are designed to fix verification errors. Otherwise, when encountering verification errors, such as "invariant not satisfied before loop" (denoted as InvFailFront) and "invariant not satisfied at end of loop body" (denoted as InvFailEnd), EXVERUS will step to counterexample generation (Section 3.1) and mutation-based counterexample-guided repair (Section 3.2).

# E. ObfsBench Dataset Construction

We curate a specialized prompt that involves few-shot examples of a set of widely-used obfuscation strategies, and prompt an LLM to generate obfuscated tasks (both verified and unverified version). In case the verified version does not pass verification, we employ an iterative repair process guided by error messages and the original proof. This process yielded a challenging but verifiable set of 266 out-of-distribution tasks.

- **Layout.** This strategy modifies the code's visual appearance and non-functional elements. E.g., *Identifier renaming* replaces descriptive variable and function names with generic or obscure identifiers to mask their intended purpose (e.g., changing `quotient` to `x`).

- **Data.** This category focuses on complicating the program's data storage and manipulation. Techniques include *Dead Variable Insertion*, which introduces variables and operations that have no effect on the final output (e.g., inserting `let mut junk = x * 3; junk = junk + 1;` where `junk` is unused). Furthermore, *Instruction Substitution* replaces simple operations with functionally equivalent, yet more complex, sequences of instructions (e.g., transforming `y = 191 - 7 * x;` into `let s = 7 * x; y = 191 - s;`).

- **Control flow.** This category alters the program's execution path, making the sequence of operations difficult to follow. Examples include *Dead Code Insertion*, which embeds blocks of code that are guaranteed never to be executed (e.g., `if (1 == 0) { y = 0; }`). Another technique is the use of *Opaque Predicates*, which are conditional expressions whose outcome is constant but is difficult for static analysis to determine (e.g., `if x * x >= 0 { ... }`). Finally, *Control Flow Flattening* disrupts structured control flow by creating redundant branches with identical operations (e.g., a redundant `if-else` structure), making the execution trace much harder to reconstruct.

# F. In-depth Analysis on Why It Is Hard to Decompile Counterexamples from Verus Backend.

Reconstructing a source-level counterexample from Verus' SMT backend is fundamentally difficult because the VC generation pipeline is intentionally *lossy*. During lowering, Verus resolves key Rust semantics (e.g., ownership, borrowing, and lifetimes) before emitting verification conditions, and compiles rich source constructs (e.g., generic collections, ghost state, and higher-level specs) into low-level SMT encodings. This translation introduces auxiliary artifacts such as SSA snapshots and internal symbols, and it erases the semantic metadata that users rely on for interpretation (e.g., high-level types, structured data layouts, and the correspondence between program variables and encoded memory). Consequently, a solver model is a valuation over these lowered artifacts rather than over a faithful source-level state; mapping it back requires recovering missing structure and aliasing/borrowing context that is no longer present, so any "decompiled" counterexample is at best heuristic and can be incomplete or misleading.

*Table 4.* Distribution of verification errors collected on VerusBench (146 tasks). Occ. is the absolute number of errors. Percentages marked with NC exclude compilation errors from the denominator.

| Error | Occ. | Occ. % | Occ. % (NC) | Steps | Step % | Step % (NC) | Tasks | Task % |
|---|---|---|---|---|---|---|---|---|
| **InvFailFront** | **754** | **26.82** | **32.81** | **419** | **32.21** | **53.17** | **86** | **58.90** |
| **InvFailEnd** | **662** | **23.55** | **28.81** | **444** | **34.13** | **56.35** | **91** | **62.33** |
| compilation_error | 513 | 18.25 | n/a | 513 | 39.43 | n/a | 143 | 97.95 |
| PostCondFail | 361 | 12.84 | 15.71 | 329 | 25.29 | 41.75 | 83 | 56.85 |
| ArithmeticFlow | 330 | 11.74 | 14.36 | 304 | 23.37 | 38.58 | 67 | 45.89 |
| PreCondFailVecLen | 137 | 4.87 | 5.96 | 98 | 7.53 | 12.44 | 29 | 19.86 |
| PreCondFail | 23 | 0.82 | 1.00 | 20 | 1.54 | 2.54 | 18 | 12.33 |
| AssertFail | 22 | 0.78 | 0.96 | 21 | 1.61 | 2.66 | 5 | 3.42 |
| Other | 7 | 0.25 | 0.30 | 7 | 0.54 | 0.89 | 1 | 0.68 |
| DecFailEnd | 2 | 0.07 | 0.09 | 2 | 0.15 | 0.25 | 1 | 0.68 |

## G. Filtering Policies

### G.1. Filtering Process for Building InvariantInjectBench

We select 142 tasks that require invariants from VerusBench, instruct the LLM to inject a high-quality and challenging one-line invariant bug using each of the three following prompts: *invariant strengthening*, *invariant weakening*, and *invariant removal*. Then we apply the following filters to get the high-quality dataset:

(1) The injected proof is buggy, leading to verification error(s) (instead of compilation error)

(2) The injected proof should contain at least one error of the expected error type, w.r.t. the prompt. For example, "invariant not satisfied at end of loop body" for *invariant weakening* and *invariant removal* injection, and "invariant not satisfied before loop" for *invariant strengthening* injection.

(3) The injected proof should only be one-invariant-different from the ground-truth proof.

After applying the above filters, we obtain 187 (out of 426) slightly buggy proofs.

### G.2. Dafny2Verus Dataset Curation

When inspecting the tasks, we find that many of the tasks show signs of reward hacking via the inclusion of tautological preconditions and postconditions that make the programs trivial to verify. This is a known problem in synthetic data generation for verification (Aggarwal et al., 2025; Xu et al., 2025). To mitigate this concern, we follow an LLM-as-judge approach similar to that of the rule-based model proposed by AlphaVerus. Given a program, we prompt an LLM to evaluate whether it contains specifications that lead to a trivial program, to decide whether the program should be rejected or not. We repeat this process five times, each with a slight prompt variation, and take a majority vote, resulting in 67 high-quality proof tasks.

## H. Extended Results on EXVERUS and AUTOVERUS

### H.1. Distribution of Different Verification Errors

To justify our focus on invariant repair, we measure the distribution of verification errors produced during repair trajectories on VerusBench. As shown in Table 4, invariant errors are the most prevalent verification failures: InvFailFront and InvFailEnd together account for 50.37% of all error occurrences and 61.62% of all non-compilation error occurrences. They also appear in a large fraction of repair steps and tasks, indicating that invariant reasoning is a central bottleneck in Verus proof generation.

### H.2. Unvalidated Counterexamples Can Still Help Proof Repair

Our validation module is deliberately conservative: it validates counterexamples for invariant errors, where the validation problem can be reduced to checking one loop transition, but it does not currently validate all verification errors. Even so,

unvalidated counterexamples can still be useful to the repair. They may not certify a real source-level failing execution, but they provide a structured hypothesis about the failing path, the relevant variables, and the missing proof fact. This can turn a broad verifier message into a concrete proof obligation for the mutator.

**Example.** In VerusBench task `mbpp_task_id_3`, the function `is_non_prime` returns `true` when it finds an `index` such that `n % index == 0`. After the first repair attempt fixed a compilation error and added loop invariants, the remaining failure was a `PostCondFail` at the early exit `return true`: Verus could not prove the postcondition `result == (exists|k:  int| 2 <= k < n && is_divisible(n as int, k))`. Since this is not an invariant error, the generated counterexamples were not validated by our invariant validation module: the trajectory records 10 generated counterexamples and `validated_true = 0`.

Nevertheless, the unvalidated counterexamples were informative. They all shared the same pattern, e.g., `n = 4, index = 2`, `n = 6, index = 2`, up to `n = 22, index = 2`. These examples point to the branch where `index` is a concrete divisor of `n`; the missing proof step is not a new invariant, but the witness needed by the existential postcondition. In other words, the examples suggest that the verifier needs help connecting the branch condition `n % index == 0` to the specification predicate `is_divisible(n as int, index as int)`.

*Listing 6.* The unvalidated counterexamples point to the missing witness for the existential postcondition.

```
1  // Before: Verus sees the early return, but cannot prove
2  // exists |k| 2 <= k < n && is_divisible(n as int, k).
3  if ((n % index) == 0) {
4      return true;
5  }
6
7  // After: the CEX pattern n = 4, index = 2 suggests
8  // that index itself should be the existential witness.
9  if ((n % index) == 0) {
10     assert(is_divisible(n as int, index as int));
11     return true;
12 }
```

Guided by this signal, the mutator produced the minimal patch shown in Listing 6, which inserts `assert(is_divisible(n as int, index as int));` immediately before `return true`. Together with the existing facts `2 <= index <= n` and the loop guard `index < n`, this assertion supplies `index` as the witness for the existential postcondition. The repaired proof then passes verification. This example illustrates that even when counterexamples are not validated, their concrete shape can still localize the logical gap and guide the LLM toward a small, semantically meaningful proof hint.

**Human-in-the-loop use.** The same signal can also be useful to a human developer writing Verus code. Instead of treating EXVERUS only as an autonomous repair agent, its counterexample generation and validation modules could be exposed as an interactive debugging aid in the Verus development loop. When Verus reports an invariant or postcondition failure, EXVERUS could present the developer with concrete assignments, the failing program point, and a validation status: validated counterexamples for invariant errors would be high-confidence explanations of why the current annotation is insufficient, while unvalidated counterexamples would be marked as hypotheses that still help identify the likely missing proof fact. The developer could then inspect these examples, decide whether the issue is a wrong invariant, a too-weak invariant, or a missing assertion/witness, and optionally apply a suggested patch. Such an integration would preserve human judgment while making verifier feedback more operational, turning abstract proof failures into concrete states that can guide the next invariant, assertion, or proof hint.

### H.3. Distribution of Repaired Proofs.

We present Venn charts on the number of fixed proofs to show how overlapped or complementary EXVERUS and AUTOVERUS are in terms of solving different tasks, shown in Figure 6 and Figure 7. While EXVERUS is broadly more capable, the two methods are also highly complementary, with each tool demonstrating unique strengths. Overall, EXVERUS uniquely solves 101 tasks that AUTOVERUS cannot, while AUTOVERUS uniquely solves 26 tasks.

Figure 7 reveals the source of these distinct capabilities. EXVERUS's unique strength is concentrated in more complex problems: It uniquely solves 63 tasks whose solutions require a high number of invariants, compared to only four for

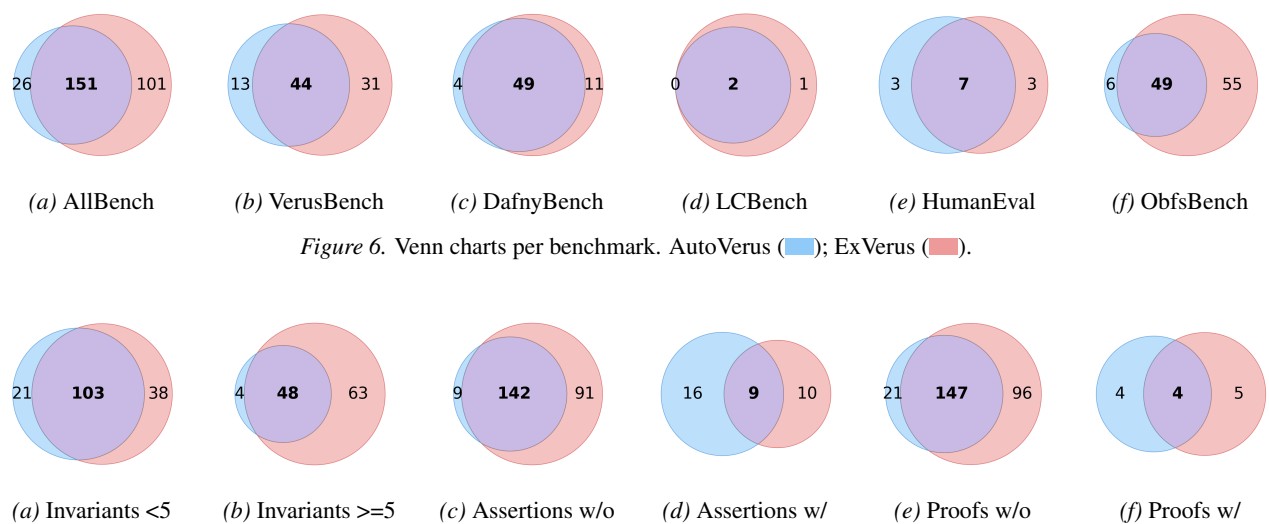

*Figure 6.* Venn charts per benchmark. AutoVerus (▮); ExVerus (▮).

*(a)* Invariants <5    *(b)* Invariants >=5    *(c)* Assertions w/o    *(d)* Assertions w/    *(e)* Proofs w/o    *(f)* Proofs w/

*Figure 7.* All-benchmarks Venn charts by difficulty (using GPT-4o). AutoVerus (▮); ExVerus (▮).

AUTOVERUS. In contrast, AUTOVERUS's unique contribution is most apparent on tasks whose solutions require the synthesis of assertions, where it uniquely solves 15 problems compared to EXVERUS's 10. But on tasks that require no assertions, it only uniquely solved 10 tasks, compared with 91 tasks solved uniquely by EXVERUS. This aligns with its design of a heuristics-based customized assertion failure repair agent, as discussed earlier. These findings again confirm EXVERUS's advantage on tasks where invariants are the bottleneck, while it is complementary to AUTOVERUS whose heuristics and heavy-weight prompting are good at repairing assertion errors.

### H.4. Performance on Tasks of Different Difficulty.

In order to compare the performance of EXVERUS and AUTOVERUS on tasks of different difficulty, we divide the tasks based on the number of invariants (low $\leq 5$ and high $> 5$), assertions (w/o and w/), and proof functions/blocks (w/o and w/) based on the ground-truth verified proofs.

To ensure a fair comparison, we normalize the ground-truth proofs before difficulty classification by pruning redundant or semantically unnecessary invariants. Therefore, we adopt a strategy inspired by the Houdini algorithm (Flanagan & Leino, 2001) to prune such invariants. Specifically, we iteratively remove each invariant and check whether its absence causes any verification errors. An invariant is deemed redundant if its removal does not affect the verification outcome. For each proof case, we enumerate invariants such as loop invariants, intermediate assertions, proof-function attributes, and proof blocks. We then comment out one component at a time, rerun Verus, and retain only those components whose absence alters the verification result. A greedy pass accumulates all redundant components, and we finally record the simplified proof corresponding to the smallest invariant set that successfully passes the verifier.

### H.5. Fine-grained Analysis on EXVERUS vs AUTOVERUS

As shown in Table 5, both with GPT-4o, EXVERUS's performance surpasses AUTOVERUS across both difficulty levels on all three difficulty dimensions on 3 out of 5 benchmarks: ObfsBench, VerusBench, and DafnyBench. This demonstrates EXVERUS generalizes across proofs with diverse categories. Though EXVERUS does not beat AUTOVERUS in some minor conditions, those marginal disadvantages do not undermine its overall superiority across the broader spectrum of tasks. On VerusBench, AUTOVERUS proves more successful on the challenging tasks that require the synthesis of assert statements (39.4% vs. 18.2%) and proof blocks (36.4% vs. 9.1%). This aligns with AUTOVERUS's design, which features a sophisticated, multi-agent debugging phase specifically engineered to generate and repair these complex proof annotations.

In fact, AUTOVERUS involves 10 dedicated repair agents for different verification errors, e.g., `PreCondFail`, `InvFail-Front`, `AssertFail`, etc.. The `AssertFail` agent will select a customized prompt based on the fine-grained error type, e.g., if the assertion error contains the keyword `.filter(`, it will use the prompt *"Please add 'reveal(Seq::filter);' at the beginning of the function where the failed assert line is located. This will help Verus understand the filter and hence*

*Table 5.* Success rate categorized by different bisections (number of invariants, w and wo assertions, w/ and wo proof functionsblocks) across different benchmarks. We use GPT-4o in this experiment.

| Benchmark | Technique | Invariants | | Assertions | | Proofs | |
|---|---|---|---|---|---|---|---|
| | | low | high | w/o | w/ | w/o | w/ |
| VerusBench | AUTOVERUS | **70.0** | 17.4 | 38.9 | **39.4** | 39.3 | **36.4** |
| | EXVERUS | 58.3 | **46.5** | **61.1** | 18.2 | **54.8** | 9.1 |
| DafnyBench | AUTOVERUS | 87.9 | 25.0 | 80.3 | **0.0** | 80.0 | **100.0** |
| | EXVERUS | **93.1** | **75.0** | **90.9** | **0.0** | **90.8** | **100.0** |
| HumanEval | AUTOVERUS | 21.4 | **3.8** | 29.4 | 9.8 | **36.4** | 4.3 |
| | EXVERUS | **23.8** | 0.0 | 29.4 | 9.8 | 27.3 | **8.7** |
| LCBench | AUTOVERUS | **28.6** | 0.0 | **50.0** | 0.0 | 33.3 | **0.0** |
| | EXVERUS | **28.6** | **4.8** | **50.0** | **4.2** | **50.0** | **0.0** |
| ObfsBench | AUTOVERUS | 35.7 | 16.1 | 19.7 | **25.0** | 20.8 | 8.3 |
| | EXVERUS | **71.4** | **30.3** | **40.6** | **25.0** | **39.6** | **25.0** |

*prove anything related to the filter.".* Such heuristics and customized prompting can help solve more tasks that require assertions/proofs, thus complementing EXVERUS whose focus is on refining invariants instead of assertions/proofs.

Additionally, on HumanEval, EXVERUS does not always outperform AUTOVERUS on more difficult tasks (>5 number of invariants) (0.0% vs. 3.8% on HumanEval) and tasks that do not require proof synthesis (27.3% vs. 36.4%). But it is noticeable that AUTOVERUS's success rate is very close to EXVERUS, which means AUTOVERUS only gains very little advantage over EXVERUS.

### H.6. AUTOVERUS Results with Different Verus Versions.

Compared to the official result of AUTOVERUS in VerusBench, there is a performance drop in the reproduction with our experiment setting, which is caused by the version of Verus. Specifically, our reproduction of AUTOVERUS with the same described setting, i.e., GPT-4o and Verus version of 2024/8/13 on VerusBench obtains a result of 75.33%, close to the reported numbers in the original paper. However, using the 2025/7/12 version, the performance degrades to 52.7%.

After investigation, there are two reasons that caused the degradation. Firstly, we found that AUTOVERUS's prompts appear to be coupled to Verus version 2024/8/13 and do not work well with the newer ones. For example, AUTOVERUS's prompts describing error fixing strategies are tailored for error messages specific to Verus version 2024/8/13, while the Verus version 2025/7/12 has different error patterns. Additionally, with the advancement of Verus, Verus version 2025/7/12 has more rigorous requirements for a proof to be verified compared to the version 2024/8/13, which means some proofs that identified as verification success by Verus version 2024/8/13 will be identified as verification failure by Verus version 2025/7/12.

```
{
    "error_category": "too_weak",
    "rationale": "The failing invariant is `exists |j: int| 0 ≤ j < i ⟹ nums@[j] == max`. This invariant is a correct
property of the intended algorithm; the variable `max` should indeed always hold a value that has been seen in the prefix
of the array `nums[0..i]`. Thus, it is not a `wrong_fact`. The error type is `InvFailEnd`, which means the verifier cannot
prove that this invariant is maintained across a loop iteration. The provided counterexamples are spurious, meaning they
represent states that are not reachable in an actual execution. For instance, the counterexample `{'__vec__nums__len': 2,
'__vec__nums__0': -1, '__vec__nums__1': -1, 'i': 1, 'max': 0}` shows `max` as `0`, a value that does not exist in the
array `[-1, -1]`. The verifier generates these spurious states because the other invariants are not strong enough to fully
constrain the relationship between `max` and the array elements. The verifier needs to be able to prove that the `exists`
invariant holds, but the current set of invariants is insufficient to eliminate these impossible scenarios. This indicates
the overall set of invariants is too weak."
}
```

*Figure 8.* A real example of error category given by the LLM-based error triage.

## I. Detailed Cost Analysis

### I.1. Cost Comparison with Iterative Refinement

EXVERUS is more expensive than Iterative Refinement on average, because it spends additional calls on counterexample generation, validation, and mutation-based repair rather than directly prompting the model with verifier errors. The key

*Table 6.* Average cost of EXVERUS on VerusBench when varying the number of generated counterexamples. Solved reports verified tasks out of 146. Each cost cell reports total / solved / failed tasks.

| Strategy | Solved | Time (s) | Tokens (K) | Money ($) |
|---|---|---|---|---|
| EXVERUS (2 counterexamples) | 95 | 1150.19 / 418.47 / 2513.21 | 97.8 / 36.0 / 212.9 | 0.025 / 0.009 / 0.054 |
| EXVERUS (3 counterexamples) | 90 | 864.60 / 414.93 / 1587.29 | 103.1 / 48.5 / 190.8 | 0.025 / 0.012 / 0.047 |
| EXVERUS (5 counterexamples) | 102 | 933.28 / 359.76 / 2262.80 | 127.9 / 45.8 / 318.2 | 0.030 / 0.011 / 0.075 |
| EXVERUS (8 counterexamples) | 98 | 1159.73 / 455.42 / 2597.71 | 121.2 / 48.6 / 269.4 | 0.027 / 0.011 / 0.060 |
| EXVERUS (10 counterexamples) | 102 | 1131.43 / 509.75 / 2572.59 | 129.6 / 57.4 / 296.9 | 0.029 / 0.013 / 0.067 |

question is whether EXVERUS's extra budget leads to more solved tasks compared to a cheaper baseline. Figure 9 (left column) answers this by plotting, for each fixed budget on the x-axis, the number of VerusBench tasks solved within that budget. We include Iterative Refinement with both 10 and 20 iterations to make sure Iterative Refinement has sufficient budget to saturate, while EXVERUS uses 10 iterations by default. The scaling curves show that Iterative Refinement saturates below EXVERUS: even when given a substantially larger token budget, the 20-iteration variant underperforms EXVERUS. The same pattern appears under dollar and wall-clock budgets, indicating that EXVERUS is more cost-effective with same budget.

### I.2. Cost Sensitivity to the Number of Counterexamples

We also evaluate how the number of generated counterexamples affects the cost of EXVERUS. Table 6 reports the number of solved tasks and the average cost over all tasks, solved tasks, and failed tasks on VerusBench when using 2, 3, 5, 8, or 10 counterexamples. The average monetary cost remains similar across these settings: using 5 counterexamples, for instance, costs $0.030 per task on average, close to $0.029 with 10 counterexamples. It also solves the same number of tasks as the default 10-counterexample setting (102 out of 146) [6], while 2, 3, and 8 counterexamples solve 95, 90, and 98 tasks, respectively. This is because multiple counterexamples are usually produced by one Z3Py query script, so increasing the requested number of counterexamples does not linearly multiply the LLM cost.

Figure 9 (right column) shows the corresponding scaling curves. Using 5 counterexamples achieves the same performance as the default 10-counterexample setting while keeping similar average cost, suggesting that only a modest number of counterexamples could be sufficient in practice. The broader trend is that adding a few counterexamples improves the signal available to mutation-based repair, while the cost remains similar because counterexample enumeration is amortized within a single generated script.

## J. Prompts

### J.1. Counterexample Query Generation

**Prompt for Compilation Error Repair**

```
Given the following Rust/Verus proof code and the verification error, write a Python script
that uses the Python Z3 API to encode constraints that capture the failing condition and
produce a concrete model (counter example).

Requirements:
- The script must `import z3` and create Z3 variables with appropriate types (Int, Bool, Arrays, etc.).
- The script must assert constraints such that `z3.check()` returns `z3.sat` when the failing
  state is possible.
- Each loop is a separate environment. Please only translate the written invariants/assertions of the loop faithfully, do not
↪  add any other constraints elsewhere, e.g., facts from preconditions unless they are explicitly stated in the loop invariants
↪  or `#[verifier::loop_isolation(false)]` is specified.
- You MUST enumerate up to {num_cex} distinct satisfying models by adding a blocking clause after each model is found, and
↪  collect them.
- The script must assign a JSON-serializable list of dicts to a global variable named `__z3_cex_results__` (each dict maps
↪  variable names to concrete values).
- Vectors (naming convention for reconstruction): To avoid name collisions, when you model a Rust Vec like `arr1: Vec<i32>` using
↪  element-wise scalars, name them with a namespace as `__vec__arr1__0`, `__vec__arr1__1`, ... (contiguously from 0). Optionally
↪  include a concrete scalar `__vec__arr1__len` giving the intended number of elements. You do not need to emit the aggregated
↪  `"arr1"` entry; the system will reconstruct `"arr1": "vec![...]"` from your namespaced entries (and `__len` if provided). If
↪  you do emit the aggregated entry, it MUST be a STRING like ``"vec![1, 2]"``.
- Keep the script minimal and concrete. Use small integer values where possible.
```

---

[6]This experiment is rerun for cost-tracking and the number is slightly lower than the one in Table 1 (105) due to randomness.

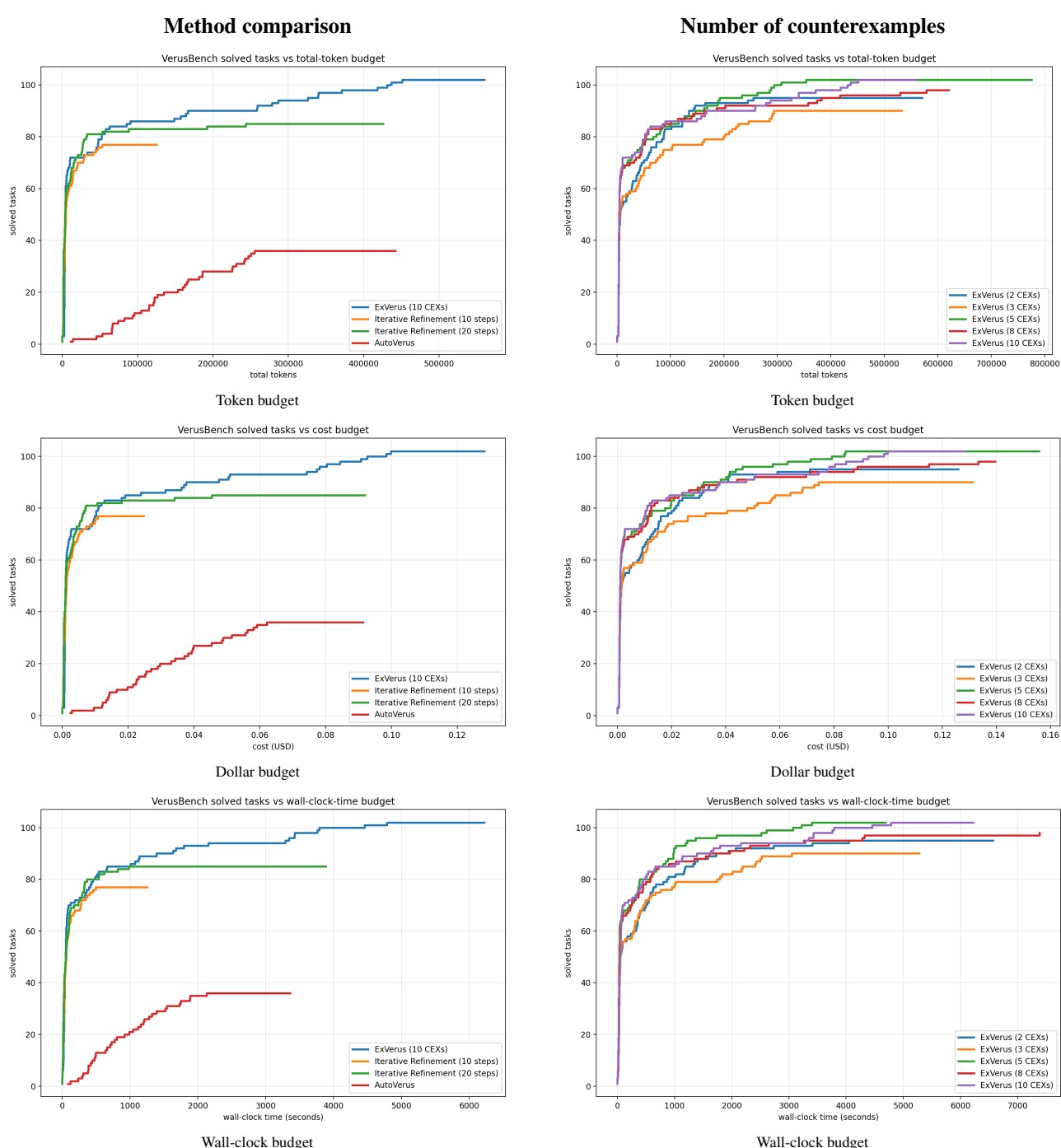

*Figure 9.* Scaling curves on VerusBench. The left column compares EXVERUS, AUTOVERUS, and Iterative Refinement; the right column compares EXVERUS with different numbers of generated counterexamples. The x-axis is the available budget, and the y-axis is the number of tasks solved within that budget. All use DeepSeek-V3.1.

- You MUST encode the values of ALL variables (including arrays or vectors) in the proof/loop/invariant into the final
  results, even if they are not used in the model solving.
- You MUST not assume anything that is not explicitly stated in the loop invariants/assertions/preconditions. If a variable is
↪  not explicitly stated in the loop invariants/assertions/preconditions, you MUST NOT assume anything about it even if there
↪  are implicit/explicit assignments to it.
- You MUST avoid using Nones in the results.

Practical guidance to avoid UNSAT and runtime errors:
- If a variable like `N`, `len`, or an index is used to size arrays or in Python `range(...)`, do NOT use symbolic Z3 Ints as
↪  Python loop bounds; instead, assign a small concrete Int (e.g., `N = z3.IntVal(2)`) and use that concrete value for any
↪  Python-side constructs.
- For vectors/arrays, you may model them with explicit small concrete elements instead of Z3 Arrays when convenient, since we
↪  only need a single concrete counterexample (e.g., set `a0, a1` as IntVals and relate them, or fix `a = [0, 1]` and express
↪  constraints on indices).
- Indices and lengths should be non-negative (>= 0). Avoid expressions that require interpreting a Z3 ArithRef as a Python
↪  integer.

Minimize constraints (prefer SAT over faithfulness when ambiguous):
- Choose ONE failing assertion/condition and encode only what is necessary to make it false.
- Use tiny bounded domains (e.g., `N = 2`, indices in {0,1}).
- You may represent `Vec<i32>` internally via namespaced scalar elements `__vec__arr1__0`, `__vec__arr1__1`, ... (optionally
↪  include `__vec__arr1__len`). The system will reconstruct an aggregated `"arr1": "vec![...]"` string from these; you do not
↪  need to emit it yourself. Legacy names like `arr1_0`/`arr1_len` are also accepted.
- Summarize loops with a few relationships rather than unrolling; avoid quantifiers.
Type modeling and ranges (MANDATORY):
- Model Rust/Verus machine integer types using Z3 Int with explicit range constraints per variable. Add these type-domain
↪  constraints in addition to the translated invariants.
- Use the following ranges (assume a 64-bit target for `usize`/`isize`). Prefer exponent form (use 2**k in Python to compute
↪  2^k):
  - bool: use Z3 Bool
  - u8: 0 <= v <= 2^8 - 1
  - u16: 0 <= v <= 2^16 - 1
  - u32: 0 <= v <= 2^32 - 1
  - u64: 0 <= v <= 2^64 - 1
  - u128: 0 <= v <= 2^128 - 1
  - i8: -(2^7) <= v <= 2^7 - 1
  - i16: -(2^15) <= v <= 2^15 - 1
  - i32: -(2^31) <= v <= 2^31 - 1
  - i64: -(2^63) <= v <= 2^63 - 1
  - i128: -(2^127) <= v <= 2^127 - 1
  - usize: 0 <= v <= 2^64 - 1 (64-bit)
  - isize: -(2^63) <= v <= 2^63 - 1 (64-bit)
  - Verus `int`: unbounded Z3 Int (no range restriction)
  - Verus `nat`: Z3 Int with v >= 0
  Note: Do not model modular wraparound; just constrain variables to these ranges unless the invariant explicitly states
↪  overflow behavior.

Additional required behavior (to make parsing robust):
- The script MUST set a global variable `__z3_cex_status__` to one of the strings: `"sat"`, `"unsat"`, or `"unknown"`.
- If `__z3_cex_status__` == "sat", the script MUST also set `__z3_cex_results__` to a JSON-serializable list of up to {num_cex}
↪  concrete variable assignments.
- Ensure that each entry in `__z3_cex_results__` includes all variables (including arrays or vectors) from the proof or target
↪  loop, regardless of their involvement in the model solving process.
- If `__z3_cex_status__` == "unsat", the script SHOULD NOT set `__z3_cex_result__` (or may set it to an explanatory string/dict).
↪  The caller will treat this as no counterexample.
- If `__z3_cex_status__` == "unknown"`, the script indicates it could not determine satisfiability.
- The script should be self-contained, import `z3`, and at the end only set these globals and exit; avoid printing extraneous
↪  text.

Rust/Verus proof code:
```rust
{proof_content}
```

{extracted_loop_section}

## Targeted Verification Error:
- **Error Type of the Targeted Error**: {verus_error.error.name}
- **Error Message of the Targeted Error**: {focused_error_text}

Full verifier console output (for context):
```
{full_error_text}
```
At the end, when counterexamples exist, set `__z3_cex_status__` = "sat"` and `__z3_cex_results__` = [ {{"x": 1, "y": 2}} ]`
↪  (example, up to {num_cex}). Ensure all values are JSON serializable.

## J.2. Compilation Error Repair

**Prompt for Compilation Error Repair**

You are an experienced Rust programmer working with the Verus verification tool. Your task is to fix compilation errors in a
↪  Verus proof file.

CRITICAL RULES - NEVER MODIFY:
1. Any execution code (logic, control flow, variables, expressions, statements)
2. Function signatures or parameters

```
3. Requires/ensures function specifications
4. Return values or types
5. NEVER use data type casts (e.g., `i as usize`, `i as int`) in loop invariants

You can ONLY:
1. Fix syntax errors
2. Fix type mismatches
3. Fix missing imports
4. Fix missing dependencies
5. Fix incorrect Verus syntax

FORBIDDEN PROOF METHODS:
- NEVER use `assume(false)` or any contradictory assumptions
- NEVER use `#[verifier(external_body)]` or similar verification-skipping attributes
- NEVER use `assume()` to bypass proof obligations

ADDITIONAL GUIDANCE:
- **Compare the buggy proof with the original unverified proof** using the provided diff (`{diff}`). Use `{original_proof}` as
↪  the canonical reference of the original source. If there is any discrepancy in executable code or specifications between
↪  `{proof_content}` and `{original_proof}`, prefer the original unverified proof and do not alter its execution logic or specs.

Here is the current proof file that has compilation errors:

{proof_content}

Also include the original, unverified proof for reference (note that the repaired proof must not change any execution code,
↪  requires/ensures function specifications, etc., of the unverified proof):
{original_proof}

Also include a unified diff showing the delta between the original unverified proof and the current proof under analysis. Use
↪  this diff to identify unintended edits to executable code or specifications:
{diff}

The compiler reported the following errors:

{error_message}

Please fix the compilation errors in the code. Focus ONLY on making the code compile - don't worry about verification errors yet.
↪  Follow these guidelines:

1. Make minimal changes necessary to fix compilation errors
2. Preserve the original proof structure and intent
3. Keep all existing specifications (requires, ensures, invariants) intact
4. Fix syntax errors, type mismatches, and other compilation issues
5. Maintain all imports and dependencies
6. Every loop must have a decreases clause (after invariants)

**ABSOLUTELY FORBIDDEN PROOF METHODS:**
- NEVER use `assume(false)` or any contradictory assumptions
- NEVER use `#[verifier(external_body)]` or similar verification-skipping attributes
- NEVER use `assume()` to bypass proof obligations
- You MUST provide genuine proofs that work with the given implementation

**CRITICAL RULE FOR FIXES: PRESERVE EVERY SINGLE CHARACTER OF ORIGINAL CODE**
You can ONLY ADD proof annotations to fix errors. You CANNOT modify, delete, or change anything that exists in the original code.
↪  The original code is read-only!

CRITICAL OUTPUT REQUIREMENT:
- You MUST output the COMPLETE, FULL Verus/Rust source file after your corrections, not a diff or snippet.
- Return one fenced code block that starts with ```rust and contains the entire file content in the end, and provide the
↪  reasoning process.
- Base your code on the given proof; preserve all existing code and specifications verbatim; only add minimal fixes.

Please generate the fixed complete Verus code:
```

## J.3. Iterative Refinement

### Prompt for Iterative Refinement

```
You are a professional Verus formal verification expert. The previously generated proof failed verification, and now you need to
↪  fix it based on the error information.

CRITICAL RULES - NEVER MODIFY:
1. Any execution code (logic, control flow, variables, expressions, statements)
2. Function signatures or parameters
3. Requires/ensures function specifications
4. Return values or types
You can ONLY:
1. Add new invariants
2. Add new assertions
3. Add new proof annotations (assert statements, lemma calls)
4. Add new ghost variables

FORBIDDEN PROOF METHODS:
- NEVER use `assume(false)` or any contradictory assumptions
```

```
- NEVER use `#[verifier(external_body)]` or similar verification-skipping attributes
- NEVER use `assume()` to bypass proof obligations

**Buggy Proof:**
```rust
<buggy_proof>
```

Also include the original, unverified proof for reference (note that the repaired proof must not change any execution code,
↪  requires/ensures function specifications, etc., of the unverified proof):
```rust
<original_proof>
```

**Verus Verification Error Message:**
```
<error_message>
```

**CRITICAL REQUIREMENT – NEVER MODIFY THE ORIGINAL CODE LOGIC**
**ABSOLUTELY FORBIDDEN DURING FIXES – VIOLATING THESE WILL RESULT IN FAILURE**
**DO NOT UNDER ANY CIRCUMSTANCES:**
1. **NEVER EVER modify, change, alter, or delete ANY original code content**
2. **NEVER modify the original requires/ensures specifications**
3. **NEVER modify comments that are part of the original code**
4. **NEVER add data type casts to variables in original code and invariants**

**ABSOLUTELY FORBIDDEN PROOF METHODS:**
- NEVER use `assume(false)` or any contradictory assumptions
- NEVER use `#[verifier(external_body)]` or similar verification-skipping attributes
- NEVER use `assume()` to bypass proof obligations
- You MUST provide genuine proofs that work with the given implementation

**CRITICAL RULE FOR FIXES: PRESERVE EVERY SINGLE CHARACTER OF ORIGINAL CODE**
You can ONLY ADD proof annotations to fix errors. You CANNOT modify, delete, or change anything that exists in the original code.
↪  The original code is read-only!

CRITICAL OUTPUT REQUIREMENT:
- You MUST output the COMPLETE, FULL Verus/Rust source file after your corrections, not a diff or snippet.
- Return exactly one fenced code block that starts with ```rust and contains the entire file content.
- Base your code on the given proof; preserve all existing code and specifications verbatim; only add minimal fixes.

Please ONLY generate the fixed complete Verus code, wrapped in the fenced code block:
```

## J.4. Mutation-based Counterexample-Guided Repair

Prompt 1: Replacing-based mutator

**Mutator Prompt (wrong fact)**

```
# Mutator: wrong_fact

Task: Remove or minimally weaken invariants/assertions that are contradicted by the counterexample(s).
Do not change executable code or requires/ensures. Keep changes minimal and sound.

CRITICAL RULES – NEVER MODIFY:
1. Any executable code (logic, control flow, variables, expressions, statements)
2. Function signatures or parameters
3. Requires/ensures function specifications
4. Return values or types
5. NEVER use data type casts (e.g., `i as usize`, `i as int`) in loop invariants
6. Never use `old` in the loop invariant

Few-shot mutations:
{examples}

Current proof:
```rust
{proof_content}
```

Inferred verdict rationale:
{verdict_rationale}

Error: {error_type} -- {error_message}
Console output:
```
{console_error_msg}
```
Counterexamples:
```
{counter_examples}
```
Original (reference, DO NOT change code/specs):
```rust
{original_proof}
```

```
```
Unified diff (reference for unintended edits):
```
{diff}
```

Output the fixed proof with updated invariants, wrapped in a single Rust block ```rust ``` in the end and a brief
↪  explanation of what you changed and why.
```

Prompt 2: Strengthen-based mutator

## Mutator Prompt (too weak)

```
# Mutator: too_weak

Task: Strengthen invariants minimally to make them inductive. Prefer semantic patterns (progress, guards, coupling) that block
↪  the CE and generalize.
Do not change executable code or requires/ensures.

CRITICAL RULES - NEVER MODIFY:
1. Any executable code (logic, control flow, variables, expressions, statements)
2. Function signatures or parameters
3. Requires/ensures function specifications
4. Return values or types
5. NEVER use data type casts (e.g., `i as usize`, `i as int`) in loop invariants
6. Never use `old` in the loop invariant

Few-shot mutations:
{examples}

Current proof:
```rust
{proof_content}
```

Inferred verdict rationale:
{verdict_rationale}

Error: {error_type} -- {error_message}
Console output:
```
{console_error_msg}
```
Counterexamples:
```
{counter_examples}
```
Original (reference, DO NOT change code/specs):
```rust
{original_proof}
```
Unified diff (reference for unintended edits):
```
{diff}
```

Output the fixed proof with updated invariants, wrapped in a single Rust block ```rust ``` in the end and a brief
↪  explanation of what you changed and why.
```

Prompt 3: Mutator for other errors

## Mutator Prompt (others)

```
# Mutator: other

Task: Make minimal, semantically meaningful invariant/assertion adjustments to address the failure while preserving behavior and
↪  specs.
Do not change executable code or requires/ensures.

CRITICAL RULES - NEVER MODIFY:
1. Any executable code (logic, control flow, variables, expressions, statements)
2. Function signatures or parameters
3. Requires/ensures function specifications
4. Return values or types
5. NEVER use data type casts (e.g., `i as usize`, `i as int`) in loop invariants
6. Never use `old` in the loop invariant

Few-shot mutations:
{examples}

Current proof:
```rust
{proof_content}
```
```

```
Inferred verdict rationale:
{verdict_rationale}

Error: {error_type} -- {error_message}
Console output:
```
{console_error_msg}
```
Counterexamples:
```
{counter_examples}
```
Original (reference, DO NOT change code/specs):
```rust
{original_proof}
```
Unified diff (reference for unintended edits):
```
{diff}
```

Output the fixed proof with updated invariants, wrapped in a single Rust block ```rust ``` in the end and a brief
↪  explanation of what you changed and why.
```

## J.5. Error Triage

### Prompt for Error Triage

```
# Verdict Inference for Invariant Repair

Classify the failure into one of: wrong_fact, too_weak, other.

Given:
- Proof:
```rust
{proof_content}
```
- Error Type: {verus_error.error.name}
- Error Message: {verus_error.get_text()}
- Console output:
```
{console_error_msg}
```

Counterexamples (if any):
```
{cex_info}
```

Please reason step by step on whether the counterexamples are reachable states or spurious states.

Domain knowledge:
- If the error is `invariant not satisfied before loop`, the invariant is likely a wrong fact and needs to be weakened or
↪  removed. Or it is missing a fact that was not explicitly stated previously, e.g., not stated in prior loops.
- If the error is `invariant not satisfied at end of loop body`, the invariant could be a wrong fact or correct but too weak;
↪  propose strengthening if plausible or replace it with a correct one.
- PreCondFailVecLen, PreCondFail, and ArithmeticFlow often indicate missing bounds over array indices or variables, suggesting
↪  the invariant is too weak.
- If all invariants are correct, the error is likely other.
- If an invariant is a correct fact but still got `invariant not satisfied before loop` error, it's possible that an dependent
↪  invariant/fact is not stated in prior loops and should be added.
- `old` is not allowed in the loop invariant.
- For errors not related to invariants or bound overflow/underflow, the error is likely other.
- For `other` error, when the invariants look correct, we likely need to add/fix some assertions to fix it.
- The provided counterexamples are not necessarily reachable states, they could be spurious states that satisfy the invariants
↪  but fail the invariants after one iteration.
- No counterexamples provided does not mean there are no counterexamples.

Instructions:
1) Decide whether the invariant/assertion is a wrong_fact, too_weak, or other. Use the knowledge above.
2) Consider CE reachability: real/reachable => wrong_fact; spurious => too_weak.
3) InvFailFront is usually wrong_fact (but not always); InvFailEnd can be either wrong_fact or too_weak.
4) PreCondFailVecLen, PreCondFail, and ArithmeticFlow usually imply too_weak (missing bounds).
5) If there are counterexamples provided, please show how counterexamples help you decide the verdict.

Output strictly as JSON:
{"verdict": "wrong_fact|too_weak|other", "rationale": "..."}
```

## J.6. Direct Proof Repair with Expert Knowledge Encoded (EXVERUSNO_MUT)

---

**Direct Proof Repair Prompt**

```
# Proof Repair Task

You need to fix the Verus verification failure by modifying invariants, assertions, or decreases clauses as needed.

## Current Proof Code:
```rust
{proof_content}
```

## Targeted Verification Error:
- **Error Type of the Targeted Error**: {error_type}
- **Error Message of the Targeted Error**: {error_message}

Full verifier console output (for context):
```
{console_error_msg}
```
{cex_info}

## Your Task:
## Repair Guidance By Error Type
### ArithmeticFlow
Fix bounds to prevent overflow/underflow. Options:
- **Add bounds**: `x <= MAX_VALUE - increment`, `x >= MIN_VALUE + decrement`
- **Fix division safety**: ensure `divisor != 0` and `divisor > 0` if needed
- **Remove overly restrictive bounds** that can't be maintained
- **Correct wrong bounds** that don't match the actual algorithm
### InvFailFront
The invariant is false when the loop starts. Options:
- **Weaken the invariant** to be true initially
- **Remove incorrect invariants** that don't hold at loop entry
- **Fix wrong conditions** in the invariant
- **Add intermediate assertions** before the loop to establish the invariant
### InvFailEnd
The invariant is not preserved by the loop body. Options:
- **Inductive strengthening** by adding a new invariant that can make the invariants preserved and inductive
- **Weaken overly strong invariants** that can't be maintained
- **Remove incorrect invariants** that don't match the loop logic
- **Fix wrong conditions** that don't account for loop body changes
- **Add intermediate assertions** to help maintain the invariant
### PostCondFail
The postcondition is not satisfied when the function returns. Options:
- **Strengthen loop invariants** to imply the postcondition
- **Remove incorrect invariants** that contradict the postcondition
- **Add bridging assertions** between invariant and postcondition
- **Fix wrong invariant conditions** that don't lead to the postcondition
### PreCondFail
A function call's precondition is not satisfied. Options:
- **Add assertions** before the function call
- **Strengthen invariants** to ensure preconditions hold
- **Remove incorrect assertions** that prevent the precondition
- **Fix wrong conditions** in invariants or assertions
### AssertFail
An assertion is failing. Options:
- **Strengthen invariants** to imply the assertion
- **Remove incorrect assertions** that don't actually hold
- **Fix wrong assertion conditions** that don't match the program logic
- **Replace assertions with weaker conditions** that do hold
- **Add intermediate assertions** to build up to the failing one
### default
Analyze the error and modify the relevant invariants or assertions as needed.
Consider strengthening, weakening, fixing, or removing conditions to make the proof work.

Also include the original, unverified proof for reference (note that the repaired proof must not change any execution code,
↪    requires/ensures function specifications, etc., of the unverified proof):
{original_proof}

Also include a unified diff showing the delta between the original unverified proof and the current proof under analysis. Use
↪    this diff to identify unintended edits to executable code or specifications:
{diff}

## CRITICAL RULES - NEVER MODIFY:
1. Any execution code (logic, control flow, variables, expressions, statements)
2. Function signatures or parameters
3. Requires/ensures function specifications
4. Return values or types
5. NEVER use data type casts (e.g., `i as usize`, `i as int`) in loop invariants

## What you CAN modify:
1. **Loop invariants** - strengthen, weaken, correct, or remove as needed
2. **Decreases clauses** - fix, add, or modify termination arguments
3. **Intermediate assertions** - add, modify, or remove helpful proof steps
4. **Proof annotations** - add, modify, or remove assert statements and lemma calls within proof blocks

## Output Requirement:
Provide the COMPLETE, FULL fixed Rust/Verus code in a single fenced code block:
```

````rust
// Your complete fixed code here
````

Then provide a brief explanation of what you changed and why.

## Best Practices:
1. **Make minimal changes** – only fix what's needed
2. **Ensure invariants are inductive** – they must be preserved by the loop body
3. **Use concrete bounds** when possible (e.g., `x <= 100` rather than complex expressions)
4. **Remove overly strong invariants** that cannot be maintained
5. **Fix incorrect assertions** that don't actually hold
6. **Ensure decreases clauses actually decrease** on each iteration
7. **Consider whether assertions should be invariants** or vice versa

Fix the proof now:

## J.7. Obfuscation

---

**Obfuscation Prompt**

### ROLE
You are an expert Rust engineer and formal-methods "obfuscator."
Your job is to make proving properties of code with Verus significantly harder, **while leaving the run-time semantics
↪ unchanged**.

### INPUT
I will paste a Rust source file.
It may include
  • ordinary Rust code,
  • verus annotations: `verus! { ... }` blocks,
  • verus annotations: specifications, i.e., preconditions `requires` and postconditions `ensures` statements,
  • verus annotations: proof annotations including invariants, `assert` and lemma functions, etc.

### TASK
Produce a *semantically equivalent* proof program that still compiles and can be verified (and, if specs are present, can still
↪ be verified with enough manual effort), but whose structure, data flow, and specs are much harder for automatic invariant
↪ generators or theorem provers to analyse (the invariants and other proof annotations should be kept or translated so that
↪ the transformed program can still be verified, and in later steps we would mask out the invariants etc.).

### EXAMPLE TRANSFORMATION IDEAS (feel free to use any combination)
* **Control-flow reshaping** – split or interleave loops; run multiple counters in opposite directions; toggle which branch
↪ executes using a flip-flop; start indices at -1 or a large offset and adjust inside the loop; add "skip" iterations.
* **State bloating** – introduce extra mutable variables (dummy accumulators, hash-like mixes, XOR chains) that never affect
↪ outputs but must be tracked in invariants.
* **Boolean camouflage** – rewrite simple conditions via De Morgan, nested implications, chained equalities, redundant
↪ inequalities, or arithmetic equivalents (`(x&1)==0` vs `x%2==0`).
* **Quantifier rewrites** – swap `forall`/`exists` with logical negation; add unused triggers; turn conjunctive predicates into
↪ implication chains.
* **Arithmetic indirection** – replace literal tables with code-point math, encode ranges via subtraction, or use non-linear
↪ equalities (`lo + hi == c`) that couple variables.
* **Dead-yet-live code** – unreachable branches that nonetheless mutate locals; checked arithmetic whose overflow path is
↪ impossible; redundant casts that blow up the type space.
* **Representation tricks** – store booleans as `u8`, counters in mixed signed/unsigned types, cast indices to wide `int` in spec
↪ contexts, pack flags into bitfields.
* **Abstraction wrappers** – hide core tests in small `const fn`, closures, or macros; inline small lambdas that reverse or
↪ double-negate results.

These are suggestions, *not* hard requirements--feel free to invent other tactics.

### OTHER NOTES
<other_notes>

### MUST-KEEP GUARANTEES
* Same observable behaviour for all inputs (return value, panics, side effects, i.e., semantics).
* No undefined behaviour or extra `unsafe`.
* Public function signatures remain intact.
* The transformed file compiles with the same toolchain; specs, if any, remain satisfiable in principle.

### OUTPUT
Reasoning process with obfuscated rust program in the end, wrapped by ```rust ```

Original program:
<ori_program>

---

## J.8. Hands Off Approaches on VeruSAGE

### J.8.1. PROMPT 1: ORIGINAL PROMPT FOR HANDS OFF APPROACH USED BY (YANG ET AL., 2025C).

---

**Prompt for Hands Off Approach**

```
The file {filename} cannot be verified by Verus, a verification tool for Rust programs, yet. Please add proof annotations to
↪  {filename} so that it can be successfully verified by Verus, and write the resulting code with proof into a new file,
↪  {output_filename}. Please invoke Verus to check the proof annotation you added. The vstd folder in the current directory is a
↪  copy of Verus' vstd definitions and helper lemmas; please feel free to check it when needed. You should KEEP editing your
↪  proof annotations until Verus shows there is no error. You should NOT change existing functions' preconditions or
↪  post-conditions; you should NOT change any executable Rust code; and you should NEVER use admit(...) or assume(...) in your
↪  code. You are also NOT allowed to create unimplemented, external-body lemma functions --- for any new lemma functions you
↪  add, you should provide complete proof. You are NOT allowed to create new axiom functions or change the pre/post conditions
↪  of existing axiom functions, and you should NEVER add external_body tag to any existing non-external-body functions. I have
↪  installed Verus locally; you can just run Verus. Before you are done, MAKE SURE to run python verus_checker.py {filename}
↪  {output_filename} to double check whether you have made any illegal changes to {filename} (fix those if you did).
```

---

### J.8.2. PROMPT 2: COUNTEREXAMPLE AUGMENTED HANDS OFF APPROACH.

---

**Prompt for Counterexample Augmented Hands Off Approach**

```
You previously attempted to verify {filename} but the verification failed. I have saved your previous attempt in {step1_output}.
↪  The verification errors from your previous attempt are in {verification_errors}. The target function to prove is usually at
↪  the end of the file.

Please analyze the verification errors and use counterexamples to fix them systematically:

APPROACH:
1. Read {verification_errors} and analyze ALL verification errors. Identify errors that represent the biggest bottleneck that
↪  you will tackle first.

2. For the error you chose to tackle, generate a counterexample in BOTH formats:
    A) Natural language explanation: Write to counterexample_1_explanation.txt
        – Describe the error in plain English
        – Explain what property is violated and why
        – Provide concrete example values that would cause the violation

    B) Concrete value assignments: Write to counterexample_1_values.txt
        – List specific values for all relevant variables
        – Show the computation that leads to the violation
        – Format: "variable_name = value" (one per line)

3. Use the counterexample to understand the root cause and fix the error in {filename}. Write your updated code to
↪  {output_filename}.

4. Run Verus to verify your fix. If this error is now resolved but other errors remain:
    – Analyze the remaining errors and choose the NEXT most important one to tackle
    – Generate counterexamples for it (counterexample_2_explanation.txt and counterexample_2_values.txt)
    – Fix that error
    – Repeat this process, strategically choosing which error to address next

5. Continue this iterative process until ALL verification errors are resolved.

6. Note that most of the required lemmas are available in the proof file, so please try to find the required lemmas based on the
↪  counterexamples, and make good use of them to fix the errors. You can search "proof fn" in the proof file to find the lemmas.
↪  You can also search "open spec" for spec functions that might be helpful (but there might be too many spec functions, so try
↪  to focus on lemmas first).

7. In intermediate steps of repairing, you can write draft solutions using unimplemented, external-body lemma functions (e.g.,
↪  admit/assume/external_body/unimplemented) to help you reason about the counterexample, verify your insights, and debug.
↪  However, in the final solution you submit in {output_filename}, MAKE SURE there is NO
↪  admit/assume/external_body/unimplemented.

IMPORTANT CONSTRAINTS:
- The vstd folder in the current directory is a copy of Verus' vstd definitions and helper lemmas; please feel free to check it
↪  when needed.
- You should KEEP editing your proof annotations until Verus shows there is no error.
- You should NOT change existing functions' preconditions or post-conditions; you should NOT change any executable Rust code;
↪  and you should NEVER use admit(...) or assume(...) in your code.
- You are also NOT allowed to create unimplemented, external-body lemma functions --- for any new lemma functions you add, you
↪  should provide complete proof.
- You are NOT allowed to create new axiom functions or change the pre/post conditions of existing axiom functions, and you
↪  should NEVER add external_body tag to any existing non-external-body functions.
- Before you are done, MAKE SURE to run python verus_checker.py {filename} {output_filename} to double check whether you have
↪  made any illegal changes to {filename} (fix those if you did).
```

---

