# OpenReview forum: "ExVerus: Verus Proof Repair via Counterexample Reasoning"
_ICML.cc/2026/Conference — ICML 2026 regular_

### Official Review · Reviewer_QAgG · 2026-03-09

**Soundness:** 2
**Presentation:** 4
**Significance:** 4
**Originality:** 3
**Overall Recommendation:** 3
**Confidence:** 4

**Summary:**

The paper studies software verification in Rust using the Verus framework. One issue there is that SMT solvers are applied to a "lowered" representation of the problem, so if the SMT solver finds a counterexample to a lowered verification condition, it is not clear how it maps back up to the original source code. The paper suggests that it is extremely hard or even technically impossible to track all metadata through the entire chain and instead produce counterexamples with LLMs.

Specifically, an iterative process is suggested where an LLM assesses the current failing correctness proof and produces a z3 script that encodes potential constraints for potential counterexamples without any "lowering". Counterexamples are automatically validated (checked whether those initial values indeed trigger assertion errors on extracted loops), categorized as loop entry or loop end failures, and passed to LLM-based proof fixing. Various counterexamples lead to various proof fix attempts which can be ranked based on the number of counterexamples they rule out and the best one will be re-used as the new starting point for proof fixing.

Comparisons to AutoVerus and standard Iterative Refinement with toolchain feedback show favorable numbers for the pipeline, while ablations assess the utility of individual parts of the pipeline.

**Compliance With Llm Reviewing Policy:**

Affirmed.

**Key Questions For Authors:**

See above, especially:
- please detail the full evaluation protocol for Iterative Refinement and ExVerus / the exact algorithm including the hyperparameters such as "required number of counterexamples".
- please report Iterative Refinement cost per model (at a minimum), at best: scaling curves for the different approaches. As mentioned, not being "uniformly best" would not weaken the paper, most likely even strengthen it.

**Limitations:**

See above, especially:
Very specific setup but one that is highly industry-relevant. Since such works have to be tailored to the specific system under consideration, using a single one should not be considered a shortcoming here.
(If proven successful, the idea of counterexample-guided reasoning in software verification will spread via followup works in other systems.)

**Strengths And Weaknesses:**

## Strengths
- Strong numbers on benchmarks indicate that the method works (provided that they are compute-matched, see below).
- Counterexample-guided reasoning is a workflow that leverages the neurosymbolic setup very well (allows LLMs to provide the parts that lower-level solvers struggle with, grounds and informs the LLMs with solver output).
- Code verification in Rust with Verus is a highly relevant application that is not far from mainstream applicability anymore.
- The paper is generally well-written and clearly introduces the problem it attempts to address.

## Weaknesses
-The benchmark evaluations lack crucial cost (compute) comparisons. The text doesn’t give enough details but it seems plausible that one iteration of ExVerus accounts for the same amount of compute as several rounds of Iterative Refinement. Specifically, the text mentions that “when EXVERUS fails to produce enough counterexamples, EXVERUS iteratively regenerates SMT queries by reflecting on the prior failures and query execution results”. The steps of error triage and mutant generation likewise require LLM usage. Some of the ablation settings seem to require lower amounts of compute and consequently perform closer to the Iterative Refinement baseline. Table 3 includes cost comparisons but crucially only for the weaker baseline and the paper’s method, omitting the Iterative Refinement baseline. Generally, performance reporting should not happen with one point (compute, performance) but with a scaling graph (by number of iterations and number of independent attempts, i.e. pass@k) cost vs. performance for the different methods. I encourage the authors to conduct this *scientific* study, not to strive for bold numbers alone. (Scaling that plateaus earlier or later, has different slopes, negative and positive results are way more useful and important findings than a method with bold numbers where it remains unclear whether it should be deployed in practice due to compute mismatched evaluations.)
- The setup is tailored to one specific shortcoming in the Verus toolchain about which the authors argue that it cannot be overcome easily. I lack the domain expertise to judge this claim independently but consider it very plausible. If it were not the case, however, the paper’s scope would be restricted to a short-term shortcoming of a domain-specific system and would benefit from being studied in other domains too.
- The paper lacks details that would enable the readers to conduct an independent cost-comparison / estimate on their own (E.g.: How many counterexamples are deemed "sufficient"? What's the exact protocol for regenerating / retrying which parts? What happens with the lower-ranked mutants?)

Overall the paper has an original and well-executed novel take on iterative invariant generation, but currently lacks the data to really back it up. Considering the many positive points, I'm willing to increase my score once accurate compute reporting and scaling has been added to the paper.

---

> ### Author Rebuttal · Authors · 2026-03-31
>
> We appreciate the reviewer for pointing out this important issue.
> We include the new experiments to more precisely compare costs.
>
> **W1 and Q2: The benchmark evaluations lack crucial cost (compute) comparisons. Please report Iterative Refinement cost per model.**
>
> We apologize for missing the fine-grained cost analysis and the IR baseline numbers.
> When fixing the number of iterations (10 iterations), the average cost per model is shown in [`this table`](https://anonymous.4open.science/r/rebuttal-65B0/cost_per_model_IR.md).
> To make sure the performance of iterative refinement (IR) saturates, we add another baseline IR(20) by extending the budget from 10 to 20 iterations.
> On average, IR is still cheaper than ExVerus using DeepSeek v3.1, e.g., 0.225x as expensive.
>
> However, we hope to emphasize that the number of iterations does not serve as a good "anchor" for comparing costs, as it does not translate into the metric we care about, such as success rate.
> To this end, the [`scaling curves`](https://anonymous.4open.science/r/rebuttal-65B0/scaling_curves_IR.md) show the success rate of different approaches using the same budget.
> When increasing the cost to around $0.05, 250K tokens, and 1000s, IR(20) tends to saturate, while ExVerus can still climb up to solve 10+ more tasks.
>
> **W2: The setup is tailored to one specific shortcoming in the Verus toolchain.**
>
> While we do not intend to overclaim that ExVerus is directly applicable to other frameworks, mapping SMT-level counterexamples back to source code is a prevalent issue across different verification frameworks.
> For example, in Dafny, the translation through the Boogie intermediate language to the Z3 solver systematically strips away structural context.
> The researchers had to engineer complex extraction tools simply to translate cryptic SMT identifiers (e.g., `T@U!val!17`) back to readable syntax [1].
> Despite the great effort invested, they had to resort to heavy heuristics, which still only support a set of simple types in Dafny and lead to spurious counterexamples due to the incompleteness of the underlying SMT solver.
> We will add the discussion to clarify our scope and the open challenges.
>
> **W3: The paper lacks details that would enable the readers to conduct an independent cost-comparison.**
>
> **Number of counterexamples:** We argued in Section 4.3 that more counterexamples contribute positively, but the discussion on how many counterexamples are sufficient is indeed missing.
> To this end, we extend the evaluation by using different numbers of counterexamples (2,3,5,8,10) and show the average cost in [`this table`](https://anonymous.4open.science/r/rebuttal-65B0/cost_diff_num_cex.md) and the trending plots in the [`scaling curves`](https://anonymous.4open.science/r/rebuttal-65B0/scaling_curves_num_CEX.md).
> Using 5 achieves the same performance as 10 for ExVerus.
> The average cost does not differ much, likely because only one Z3Py script is required to generate multiple counterexamples.
>
> **Regenerating/retrying:** ExVerus 1) reflects and regenerates Z3Py script when not getting sufficient counterexamples (less than half of expected), max_try = 5; 2) reruns one iteration if the last iteration fails, num_iteration = 10.
>
> **Lower-ranked mutants:** In the current design, ExVerus operates as a greedy search: only the top-ranked mutant is advanced to the next round, while other mutants are discarded.
> However, it can be extended to support other search strategies [2] by keeping and exploring the top-K (K > 1) candidates.
> It is indeed an interesting direction to explore in future work.
>
>
>
> **Q1: please detail the full evaluation protocol for IR and ExVerus**
>
> IR follows a feedback-driven loop.
> In each iteration, the LLM is prompted with the unverified code, the corresponding error message from Verus, and a dedicated repair prompt (Appendix J.3).
>
> ExVerus follows the detailed pipeline outlined in the pseudo-code in Appendix C (Algorithms 1/2/3).
>
> Counterexample Generation: ExVerus aims to generate 10 counterexamples per iteration. However, the experiment on W3 show that 5 can be sufficient.
>
> Mutation and Ranking:
> For each repair step, the LLM generates up to 5 mutated proof candidates.
> These candidates are ranked based on the number of blocked counterexamples. The top-ranked mutant is selected for the next iteration.
>
> Shared Hyperparameters:
>
> We follow AutoVerus by setting the temperature to 1.0 and the max_token to be 4096 for non-reasoning models. For the reasoning model o4-mini, we extend it to 40960 for all techniques due to the large amount of thinking tokens. Claude-sonnet-4.5 still uses 4096 since we use the non-thinking mode.
>
> [1] Chakarov, Aleksandar, et al. "Better counterexamples for Dafny." International Conference on Tools and Algorithms for the Construction and Analysis of Systems. Cham: Springer International Publishing, 2022.
>
> [2] Li, Dacheng, et al. "S*: Test time scaling for code generation." arXiv preprint arXiv:2502.14382 1.2 (2025).

---

### Official Review · Reviewer_tNqF · 2026-03-12

**Soundness:** 4
**Presentation:** 4
**Significance:** 4
**Originality:** 3
**Overall Recommendation:** 5
**Confidence:** 3

**Summary:**

This paper proposes a framework, ExVerus, that enables LLMs to generate and repair proofs in Verus. ExVerus consists of two core components: (1) counterexample generation with validation that leverages LLMs to synthesize source-level SMT queries that produce multiple counterexamples, followed by a designed verifier-based counterexample validation module; (2) mutation-based counterexample-guided repair that leverages LLMs to classify errors, applying customized mutation and ranking. The method is evaluated on a constructed benchmark that involves four datasets. Multiple LLMs are applied to ExVerus, and improvements are generally observed. The method is also proven to be more economical than the compared baselines. The authors further construct two benchmarks, ObfsBench and InvariantInjectBench, which demonstrate robustness to fixing obfuscating proofs and analyze the impact of the number of counterexamples, respectively.

**Compliance With Llm Reviewing Policy:**

Affirmed.

**Final Justification:**

The method is effective. The paper is solid. The rebuttal answers my questions. I recommend acceptance of this paper.

**Key Questions For Authors:**

* How does MutatorSelect() works? Is it by LLM or from Verus?

**Limitations:**

Yes

**Strengths And Weaknesses:**

Strengths:
* The proposed ExVerus shows a significantly improved success rate in multiple datasets. The evaluation is comprehensive over multiple datasets and LLMs.
* The experiments and analyses are comprehensive. The method is proved to be robust and economical.
* The paper is well-organized and easy to follow.

Weaknesses:
* It is shown in Table 4 that ExVerus_NO_MUT performs generally inferior to the compared Iterative Refinement. Explanations should be included in the paper.
* The caption of Figure 2 is uninformative. A more detailed description can help with the understanding of the workflow.
* ClaudeSonnet-4.5, GPT-4o, O4-mini, Qwen3 Coder (Qwen3480B-A35B), and DeepSeek-V3.1 are not properly cited.

---

> ### Author Rebuttal · Authors · 2026-03-31
>
> **W1: ExVerus_NO_MUT performs generally inferior to the compared Iterative Refinement. Explanations should be included in the paper.**
>
> Thank you for catching this. We examined Table 4 and found that ExVerus_NO_MUT underperforms Iterative Refinement on 2, 1, 3, 1, and 1 models (out of 5 models in total) across 5 benchmarks, respectively, i.e., underperforms on 8 out of 25 model-benchmark combinations. We attribute the performance drop (compared with ExVerus) to the critical counterexample-guided mutation and validation modules. The mutation module will select the strengthen-based mutator or the replace-based mutator based on whether the error is the invariant/assertion is "correct but too weak" or "incorrect". We will add a detailed explanation to the ablations in Section 4.2.
>
> **W2: The caption of Figure 2 is uninformative. A more detailed description can help with the understanding of the workflow.**
>
> Thank you for noting this. We will add a more informative description to the figure caption in the revised version:
>
> *The ExVerus Workflow. The framework operates in two core phases after the initial proof generation module. First, the Counterexample Generation and Validation module leverages LLMs to synthesize source-level Z3Py queries and execute them to produce concrete counterexamples, which are subsequently filtered by a dedicated verifier-based validation module. After that, the Mutation-Based Repair module uses these validated counterexamples to guide error classification, apply customized mutations, and rank the generated candidates to repair the proof.*
>
>
> **W3: ClaudeSonnet-4.5, GPT-4o, O4-mini, Qwen3 Coder (Qwen3480B-A35B), and DeepSeek-V3.1 are not properly cited.**
>
> Thank you for catching this issue. We will cite the models in the revised version.
>
> **Q1: How does MutatorSelect() works? Is it by LLM or from Verus?**
>
> `MutatorSelect()` is a rule-based mutator selector.
> When a verification failure occurs, ExVerus queries an LLM with the buggy proof, the counterexamples, and verifier feedback to categorize the error.
> The triage analyzes the characteristics of the counterexamples. MutatorSelect() selects a mutator purely based on the verdict generated by the triage (which is LLM-based). If the counterexamples are reachable from a valid initial state, the invariant/assertion is likely incorrect.
> The triage infers a verdict “wrong_fact” and the MutatorSelect() module will select a replace-based mutator to make the invariant/assertion compliant with the states.
> If the counterexamples are unreachable, spurious states that should have been filtered out by the invariant/assertion but have not yet, the invariant/assertion is likely correct but too weak.
> The triage infers a verdict “too_weak” and the MutatorSelect() module will select a strengthen-based mutator to make sure the states are filtered out by the invariant/assertion.
>
> Figure 8 and Section J.5 show the triage output example and prompt. We will include this description in the implementation details of the revised version.

---

> > ### Author Rebuttal · Reviewer_tNqF · 2026-04-03
> >
> > Thanks for the answers. My concerns are resolved.

---

> > > ### Author Response · Authors · 2026-04-03
> > >
> > > Thank you for your careful review and valuable suggestions. Your comments have helped us improve the paper by claryfiying details of ablation experiments, mutation methodologies, and adding more details about the workflow. We sincerely appreciate your supportive assessment, and we believe these revisions have made the paper clearer, more rigorous, and easier to follow.

---

### Official Review · Reviewer_AEbM · 2026-03-13

**Soundness:** 3
**Presentation:** 4
**Significance:** 3
**Originality:** 4
**Overall Recommendation:** 5
**Confidence:** 4

**Summary:**

ExVerus is a new approach to generate counterexamples that help an LLM repair a proof. Rather than rely on Verus' internal representation, it generates counterexamples based on the source code directly.

**Compliance With Llm Reviewing Policy:**

Affirmed.

**Final Justification:**

I recommend accepting this very strong paper.

**Key Questions For Authors:**

No questions.

**Limitations:**

Yes

**Strengths And Weaknesses:**

This is a fairly strong paper.

*Strengths*
- The writing is clear and this paper was very pleasant to read.
- ExVerus is clearly novel and contributes towards solving an important open problem in the field.
- The evaluation convinced me (multiple large datasets, fairly big improvements).
- Clear and transparent discussion of limitations.

*Weaknesses*
- The paper does not go into whether ExVerus is particularly good (or particularly bad) at certain types of Rust code. For instance, I would imagine that code with lots of generics would be much harder.
- I would have liked to see a discussion on whether ExVerus can be useful to a human writing Verus code. It appears that they would equally benefit from it.

---

> ### Author Rebuttal · Authors · 2026-03-31
>
> We thank the reviewer for the encouraging comments and appreciate for pointing out the issues.
>
> **W1: The paper does not go into whether ExVerus is particularly good (or particularly bad) at certain types of Rust code.**
>
> To evaluate which types of tasks ExVerus is good and bad at, we presented experiments and analyses in Appendices I.2 and I.3 (see Figures 6 and 7 and Table 5). We classify tasks by the number of invariants, assertions, and proof functions/blocks. We find that ExVerus performs better on tasks that require more invariants (>=5) and tasks that require no assertions. ExVerus underperforms AutoVerus on tasks that require assertions and proof functions, as AutoVerus has more customized, heuristics-based prompts and designs that target assertions and proof functions. We will add a forward reference in the evaluation section to refer to this analysis.
>
>
> **W2: I would have liked to see a discussion on whether ExVerus can be useful to a human writing Verus code.**
>
> We agree that ExVerus can potentially benefit developers for Verus debugging. The primary bottleneck that ExVerus resolves, i.e., generating readable and actionable counterexamples, is the exact same issue that impedes human developers from debugging Verus proofs [1]. Verification developers often get frustrated by the cryptic error messages [2], complaining that the verifier just “spits out a no” or provides “no help” [3]. The developers noted that distinguishing genuine counterexamples from solver noise requires deep experience, severely hindering productivity.
>
> By synthesizing source-level counterexamples, ExVerus can provide human developers with concrete, readable program states that reveal the verification error. This can effectively replace the tedious process of manually deciphering messy SMT-LIB models. In doing so, ExVerus could accelerate manual proof repair and lower the bar to write formally verified Rust code.
>
> We will discuss how the ExVerus counterexample generation and validation modules could be integrated into a human-in-the-loop developing process in the revised version.
>
> [1] Ayaziová, Paulína, et al. "Software verification witnesses 2.0." International Symposium on Model Checking Software. Cham: Springer Nature Switzerland, 2024.
>
> [2] Brugger, Lea Salome, Xavier Denis, and Peter MÃžller. "Toward Practical Deductive Verification: Insights from a Qualitative Survey in Industry and Academia." arXiv preprint arXiv:2510.20514 (2025).
>
> [3] Mugnier, Eric, et al. "On the Impact of Formal Verification on Software Development." Proceedings of the ACM on Programming Languages 9.OOPSLA2 (2025): 3642-3668.

---

> > ### Author Rebuttal · Reviewer_AEbM · 2026-03-31
> >
> > I thank the authors for the response. I am keeping my (positive) score.

---

> > > ### Author Response · Authors · 2026-04-03
> > >
> > > We sincerely thank the reviewer for their thoughtful feedback, constructive comments and the positive assessment of our work. We are glad to know that the concerns have been addressed.

---

### Official Review · Reviewer_7aeb · 2026-03-13

**Soundness:** 3
**Presentation:** 4
**Significance:** 3
**Originality:** 4
**Overall Recommendation:** 5
**Confidence:** 4

**Summary:**

This paper presents **EXVERUS**, an LLM-guided framework for automated proof repair in Verus (a Rust verification toolchain). It generates concrete counterexamples by prompting LLMs to synthesize source-level Z3Py queries. These counterexamples are validated for loop invariant errors and used to guide mutation-based repairs that block the failing states. Evaluated on five benchmarks, EXVERUS outperforms the state-of-the-art AutoVerus by 38% in success rates (up to 2× on complex tasks with ≥5 invariants), while reducing costs by 4.25× and runtime by 4×.

**Compliance With Llm Reviewing Policy:**

Affirmed.

**Final Justification:**

Thanks the authors for the response. I am keeping my score.

**Key Questions For Authors:**

1. In the automatically generated Z3Py scripts, how does EXVERUS ensure that the LLM captures the most critical counterexamples rather than trivial or insignificant edge cases? Is there a mechanism to automatically filter out invalid or spurious counterexamples that arise from the LLM's misunderstanding of Rust semantics?

2. EXVERUS currently focuses on repairing loop invariants and assertions. For complex verification failures involving Rust's unique Ownership and Lifetimes, is the LLM capable of generating sufficiently precise counterexamples to assist in the repair process?

3. The experiments used ObfsBench to test robustness. However, can EXVERUS maintain its performance and cost advantages when handling verification failures in large-scale industrial codebases?

**Limitations:**

yes

**Strengths And Weaknesses:**

## Strengths
- Novel bypass: Avoids the undecidable decompilation of low-level SMT artifacts by synthesizing high-level Z3Py scripts directly.
- Validated feedback: Dedicated validation module distinguishes real vs. spurious counterexamples for invariant errors, grounding LLM reasoning in concrete program states.
- Robustness: Introduces ObfsBench to show resilience against code obfuscation, mitigating concerns about LLM memorization.

## Weaknesses
- While the paper acknowledges this in Appendix B, the validation module is "currently restricted to invariant errors". Assertion failures, postcondition violations, and precondition failures rely on unvalidated CEXs, which limits the approach's completeness.

- The paper notes that "correctness of counterexamples can occasionally be unverifiable, e.g., for non-inductive cases and sophisticated invariants" (Section 2.3). However, it does not quantify how often this occurs or how the system degrades when CEX validation fails.

---

> ### Author Rebuttal · Authors · 2026-03-31
>
> **W1: The validation module is "currently restricted to invariant errors".**
>
> ExVerus only supports validating the counterexamples for invariant errors, even though it still leverages unvalidated counterexamples to repair arbitrary proof errors.
> In this case, the counterexamples fall back to structured reasoning steps like CoT.
> Empirically, this is still helpful for the reasoning process [1], e.g., [`this case`](https://anonymous.4open.science/r/rebuttal-65B0/unvalidated_counterexample_case_analysis/README.md).
>
> This [`table`](https://anonymous.4open.science/r/rebuttal-65B0/error_stats.md) shows that invariant errors account for the majority of verification errors, around 50% of total errors and 60% of verification errors (excluding compilation errors).
>
> While we did try to develop counterexample validation for other errors, we realized that some errors can be extremely challenging, and even infeasible [2]. We will add the discussion to the revised paper.
>
> **W2: The paper does not quantify how often unverifiable counterexamples occurs or how the system degrades when CEX validation fails.**
>
> Quantifying the rate of unvalidated counterexamples can help us better understand ExVerus's robustness.
> We analyzed the ExVerus trajectories and found ExVerus validated 2,261 out of 3,292 counterexamples, yielding a failing rate of 31.32%.
> In 11.32% steps, none of the counterexamples pass validation.
>
> We have to admit it is hard to quantify how the system degrades when CEX validation fails in ExVerus.
> The inference could involve multiple steps, each with a different number of validated counterexamples.
> As discussed above, we empirically observe that ExVerus still enjoys better performance with these structured reasoning steps.
> We will add more studies on the effectiveness of unvalidated counterexamples.
>
>
> **Q1: In the automatically generated Z3Py scripts, how does EXVERUS ensure that the LLM captures the most critical counterexamples rather than trivial or insignificant edge cases?**
>
> It is possible that Z3Py encounters an error or produces no counterexamples, so we have a retry mechanism that asks the LLM to reflect and regenerate counterexamples.
> Our script queries the solver in a loop, where each iteration excludes the counterexample generated in the previous iteration to ensure it is not repeated, yielding a (ideally) diverse set that supports better generalization during repair.
> Other than this, we do not explicitly check if the counterexamples are trivial, and we resort to mutation-based repair to generalize the proof from them.
>
> **Q2: For complex verification failures, is the LLM capable of generating sufficiently precise counterexamples to assist in the repair process?**
>
> For Rust exec code alone, the lifetime and borrow-checker errors are caught by the Rustc compiler before reaching the SMT solver.
> As the proof always starts from a **correct** program, there are no such errors in the execution code.
> We expect that generating precise source-level counterexamples for these failures would be challenging and likely beyond ExVerus's current capability, as it would require the LLM to accurately encode the reasoning about resource ownership and exclusivity (typically formalized as separation logic) into Z3 constraints.
> We will clarify this limitation in the discussion section.
>
> **Q3: Can EXVERUS maintain its performance and cost advantages when handling verification failures in large-scale industrial codebases?**
>
> ExVerus' counterexample validation can face challenges in large-scale industrial codebases due to long cross-file context, complex Verus macros and grammar sugars, etc.
> In fact, VeruSage-Bench [3], a new large-scale industrial Verus benchmark, shows that proofs involving invariants are less common than in smaller benchmarks evaluated in our paper.
>
> While we do not indend to overclaim about the advantages on industrial codebases, we added preliminary results by incorporating ExVerus (for generating unvalidated counterexamples) into the same agentic framework following [3], i.e., GitHub Copilot CLI based on claude-sonnet-4.5.
> We find that ExVerus can solve one challenging system-level proof synthesis task in VeruSage-Bench, which the agentic framework originally failed to solve.
> Using counterexamples costs 5.56% less input tokens and 18.70% less output tokens.
> We detail the discussion on this interesting case study [`here`](https://anonymous.4open.science/r/rebuttal-65B0/verusage_case_analysis/README.md), but we admit there remains significant room for improvement.
>
> [1] Chen, Xinyun, et al. "Teaching large language models to self-debug." arXiv preprint arXiv:2304.05128 (2023).
>
> [2] Clarke, Edmund, et al. "Counterexample-guided abstraction refinement." International Conference on Computer Aided Verification. Berlin, Heidelberg: Springer Berlin Heidelberg, 2000.
>
> [3] Yang, Chenyuan, et al. "VeruSAGE: A Study of Agent-Based Verification for Rust Systems." arXiv preprint arXiv:2512.18436 (2025).

---

> > ### Author Rebuttal · Reviewer_7aeb · 2026-04-03
> >
> > Thanks the authors for the response. I am keeping my score.

---

> > > ### Author Response · Authors · 2026-04-03
> > >
> > > Thank you for your constructive comments and valuable suggestions. Your comments have helped us improve the paper by clarifying the boundary and scope of our framework, showing how often unveriable counterexamples occurs, and demonstrating some potential in adopting the idea to repository-level tasks. We sincerely appreciate your supportive assessment, and we believe these comments have made the paper clearer, more rigorous, and easier to follow.

---

### Decision · Program_Chairs · 2026-04-30

**Decision:**

Accept (regular)

**Comment:**

This paper introduces EXVERUS, a framework for automated repairs of formal proofs in Verus, which is a verification toolchain for Rust. The key idea of it is to use LLMs to synthesize high-level Z3Py queries from the source code directly, so that it can bypass the problem where SMT-level counterexamples become unreadable. These counterexamples are then validated and used to guide a mutation-based repair progress. Multiple benchmarks, including two new ones that are constructed in this paper (ObfsBench and InvariantInjectBench) are used to show that EXVERUS outperforms pervious methods such as AutoVerus.

Overall, the reviewers all agree on the novelty of the proposed neuro-symbolic approach to use LLMs to generate the Z3Py directly, and the empirical results are quite strong, thus I'm recommending an acceptance for this paper. At the same time, the authors should try to incorporate the discussions and clarifications from the discussion period into the final version of the paper.